# Impacts of convection, chemistry, and forest clearing on biogenic volatile organic compounds over the Amazon

Nidhi Tripathi [1] ✉, Bianca E. Krumm [1], Achim Edtbauer [1], Akima Ringsdorf[1], Nijing Wang [1], Matthias Kohl [1], Ryan Vella [1,2], Luiz A. T. Machado [3], Andrea Pozzer [1,4], Jos Lelieveld [1,4] & Jonathan Williams [1,4] ✉

The Amazon rainforest is the largest source of biogenic volatile organic compounds (BVOCs) to the atmosphere. To understand the distribution and chemistry of BVOCs, airborne and ground-based measurements of BVOCs are conducted over the Amazon rainforest in the CAFE-Brazil campaign (December 2022–January 2023), including diel (24-hour) profiles between 0.3-14 km for isoprene, its oxidation products, and total monoterpenes. Although daytime deep convective transport of BVOCs is rendered ineffective by photochemical loss, nocturnal deep-convection exports considerable BVOC quantities to high altitudes, extending the chemical influence of the rainforest to the upper troposphere, and priming it for rapid organic photochemistry and particle formation at dawn. After contrasting pristine and deforested areas, a BVOC sensitivity analysis is performed using a chemistry-climate model. Here we show that reducing BVOC emissions, decreased upper tropospheric ozone, increased lower tropospheric hydroxyl radicals, shortened the methane lifetime, with the net effect of enhancing climate warming through ozone and aerosols.

Emissions of biogenic volatile organic compound (BVOC) from the Amazon rainforest occur through diverse processes such as photosynthesis, respiration, and stress responses of the vegetation, but also from microbial activities in the forest soil[1–6]. The main primary BVOC emissions are isoprene ($C_5H_8$) and monoterpenes ($C_{10}H_{16}$) with estimated Amazonian emissions of 150 Tg y$^{-1}$ and 60 Tg y$^{-1}$, respectively[7]. Once released, BVOCs are oxidized rapidly by OH radicals, ozone ($O_3$), and, to a lesser extent, $NO_3$ radicals, into multiple secondary volatile oxidized products and aerosol particles. Therefore, BVOC emissions can influence the atmospheric oxidation capacity, the radiative budget, and regional precipitation by generating secondary organic aerosols and cloud condensation nuclei, respectively[8–10]. Due to the short atmospheric lifetimes of most BVOCs (minutes-hours), their influence is generally thought to be primarily confined to the boundary layer. Indeed, a strong concentration gradient has been observed by day at the entrainment zone between the top of the planetary boundary layer (0-2.5 km) and the free tropical troposphere (2.5-18 km)[11]. Nevertheless, late afternoon shallow cumulus lift BVOC emissions to around 3 km before the onset of the nocturnal boundary layer at 100-200 m, which strands them aloft overnight in the so-called residual layer[12]. Here, they can persist since the photochemically generated OH radicals that initiate oxidation by day are negligible at night, and their reaction with $O_3$ is relatively slow. However, frequent deep convection events impact the entire Amazon region, which can potentially transport boundary layer and residual layer air up to high altitudes within 0.5-2 hours[13,14]. Substantial nocturnal vertical transport to mid-altitudes (6-8 km) has been proposed based on model and satellite data assessments of aerosol distributions, although this has

[1]Department of Atmospheric Chemistry, Max Planck Institute for Chemistry, Mainz, Germany. [2]Institute for Atmospheric Physics, Johannes Gutenberg University Mainz, Mainz, Germany. [3]Institute of Physics, University of Sao Paulo, Sao Paulo, Brazil. [4]Climate and Atmosphere Research Center, The Cyprus Institute, 1645 Nicosia, Cyprus. ✉e-mail: n.tripathi@mpic.de; Jonathan.Williams@mpic.de

not been confirmed with in-situ data[15]. A recent case study from the same Chemistry of the Atmosphere Field Experiment, Brazil (CAFE-Brazil) campaign and associated modeling have shown that prodigious particle production occurs just after dawn at much higher altitudes (12-14 km) and is related to the oxidation of isoprene[16,17]. Despite evidence for potentially profound atmospheric effects of BVOC on the oxidation capacity, particle production, and precipitation, a full spatial and temporal assessment of these BVOCs above their largest global terrestrial source is yet to be determined.

Globally, the emission of BVOCs from terrestrial vegetation is estimated to be ~760 Tg C yr$^{-1}$, of which isoprene and monoterpene contributions are 70% and 11%, respectively[3,18]. The Amazon accounts for 40% of all BVOC emissions[3] and is therefore a significant component of the global carbon cycle. In the past few decades, the Amazon rainforest has experienced significant changes driven by human activities, including deforestation, climate change, and natural disturbances like El-Nino. The Amazonian deforestation is still increasing due to human activities, notably by conversion of forested land for agriculture (cropland, pasture, and plantation) and logging[19–21]. Wright et al. (2017)[22] reported that the dry season had been extended by deforestation in the southern Amazon, and Leite-Filho et al. (2021)[23] suggested that rainfall reductions could be attributed to deforestation through analysis of historical records, thus also connecting BVOC emissions to the hydrological cycle. It is estimated that since the 1960s, approximately 20% of the rainforest has been converted to farmland, resulting in a net release of $CO_2$. In addition to deforestation, climate change has emerged as another major driver of transformation in the Amazon. Global warming has increased tree mortality rates, forest fires and exacerbated drought or water stress across the Amazon rainforest[24,25]. These factors may lead to variations in BVOC emissions, either increasing or decreasing them. To assess the full atmospheric effects of deforestation and climate change, the chemical and physical impacts of BVOC must be characterized and understood[26,27].

The main objective of the CAFE-Brazil campaign was to investigate the distribution and atmospheric processing of biogenic chemicals, as well as their role in upper tropospheric chemistry following convective transport over the Amazon rainforest during the dry to wet transition periods. Although a previous study reported the strong seasonal variation in the BVOC emissions at the surface over the Amazon rainforest[28], Machado et al. (2004)[29] reported the strong intensity of convection during the dry-to-wet transition periods and the onset of the wet season. This campaign primarily focused on BVOC measurements over pristine areas to minimize the influence of urban plumes and biomass burning (which was less prevalent during this period)[30–32]. Previous studies have shown that urban plumes can significantly influence photochemistry and the oxidation capacity in the downwind forest regions[33,34]. However, the low median value of benzene (24 ppt) in the boundary layer measured near the limit of detection (LOD, 12 to 22 ppt for different flights) further indicates that urban emissions were not a significant source in the area covered during this study (Supplementary Fig. 1).

By exploiting the extended range (8000 km), high altitude (14 km) and long flight duration (10 hours) capability of the *High Altitude and LOng-range* (HALO) research aircraft, data coverage extended between 4.5° N to 11.5° S and 33° to 72° W, see Fig. 1. By varying the take-off times of the 15 research flights, full 24-hour profiles of BVOC mixing ratios could be characterized at multiple altitudes from the campaign data. In addition to the airborne data, measurements from the Amazon Tall Tower Observatory (ATTO) ground site, situated in the pristine rainforest 150 km northeast of Manaus (2.1° S, 59.0° W), allowed 24-hour data coverage at 320 m over the two-month CAFE-Brazil campaign. The combination of airborne and ground-based datasets gives an unprecedented view of the vertical (0.3–14 km), temporal (24-hour profiles at multiple altitudes) and spatial distribution of BVOCs over the Amazon rainforest. Here, we focus on isoprene, its combined

oxidation products methyl vinyl ketone, methacrolein, isoprene-hydroxy-hydroperoxides, (hereafter termed isoprene-OP), and mono-terpenes. These data were measured using two near identical Proton-Transfer-Reaction Time Of Flight Mass Spectrometers (PTR-TOF-MS), one at the ATTO ground site and one operated in the aircraft. The instruments were cross-calibrated and validated with a Gas chromatography-mass spectrometer (GC-MS) (see methods). A low-altitude flight (900 m) contrasting pristine forested and deforested areas provided us an additional motivation to perform a BVOC sensitivity study using the global atmospheric chemistry model EMAC. This study aims to investigate the atmospheric impacts of the Amazon rainforest's current BVOC emissions and explores impacts of potential changes of these emissions. The model was then used to assess the atmospheric chemistry of BVOCs over the Amazon rainforest and the impact of changes associated with their modulation.

## Result and discussion
### Vertical profiles of trace gases

The vertical profiles of isoprene, isoprene OP, and monoterpenes for all flights are shown in Fig. 2 and Supplementary Fig. 2. The vertical distributions of these gases show similar characteristics, an approximately exponential decrease from 0.3 to 3 km in the lower troposphere (LT), a central mid-troposphere section from 3-9 km with consistently low mixing ratios, and then increased levels at higher altitudes in the upper troposphere (UT) peaking between 9 to 14 km. The higher mixing ratios at lower altitudes result from direct emissions (isoprene and monoterpenes) by the rainforest ecosystem into the well-mixed boundary layer and subsequent rapid oxidation to isoprene-OP. The primary VOCs (isoprene and monoterpenes) declined more rapidly than isoprene-OP with increasing altitude in the LT, in accord with their respective reaction rates with OH. Within the LT, mixing ratios decrease rapidly with height through both turbulent mixing dilution from above and chemical loss (oxidation). The mean mixing ratios of isoprene (2.25 ± 1.35 ppbv) and monoterpenes (0.23 ± 0.14 ppbv) between 0.3 to 1 km are ~2.5 times higher than the mean value measured between 1 to 2 km altitude. The isoprene-OP (1.51 ± 0.77 ppbv) decline less steeply to 0.87 ± 0.59 ppbv over the same height due to its secondary production from isoprene photo-oxidation during upward transport and subsequent advection aloft. In the mid-troposphere, the mean mixing ratios of all BVOCs are very low, and the levels are generally constant except for a few outliers. Such short spike-like events may be due to outflow from occasional cloud convection to this height or mid-level detrainment from deeper convective systems[35]. However, the predominately low mixing ratios in this altitude range suggest such events are rare.

The UT data showed elevated levels of isoprene, isoprene-OP and monoterpenes, with mixing ratios peaking between 11 and 12.5 km. The elevated mixing ratios of isoprene, isoprene-OP and monoterpenes in the UT provide clear evidence of extensive rapid transport to this altitude associated with deep convection. Supplementary Fig. 3 depicts an example of the deep convection of isoprene in the UT for Flight 10. The aircraft sought and captured multiple deep convection outflow events in the UT during day and night, although mixing ratios for all three species at ~12.5 km were consistently higher at night (see inset Fig. 2 and Supplementary Fig. 4). The detrainment of trace gases from convective clouds is known to depend on various parameters, including the lapse rate minimum, which determines the height of the horizontal outflow of the air emerging from the convective cloud. Gettelman and Forster[36] reported a lapse rate minimum height of 10–12 km for active convective (deep convection) regions in the tropics, in good agreement with the observed higher BVOCs at this altitude range presented here. In the present study, on average, approximately 20% of the boundary layer isoprene reaches the UT during the convective outflow, while Bardakov et al. estimated from model calculations that ~30% of the boundary layer isoprene would be

transported to the UT during an hour of convection. Jo et al.[37] measured on average ~25% of isoprene from the boundary layer was transported to the UT during the SEAC4RS campaign over the southeastern US.

Relative to the boundary layer, the mixing ratios of isoprene, isoprene-OP and monoterpenes measured after deep convective transport are significantly lower. This can result from: oxidation reactions with OH and $O_3$; partial uptake into the cloud for the more soluble isoprene-OP; or dilution through the entrainment of mid-tropospheric air masses. These processes also contribute to the large variability of BVOC mixing ratios observed in the UT, and the stochastic nature of the convective events. Significantly, the nighttime profiles show a pronounced enhancement in the mixing ratios of isoprene (0.57 ± 0.36 ppbv), its oxidation product (0.29 ± 0.22 ppbv) and monoterpenes (0.053 ± 0.015 ppbv) in the UT between 12 to 13 km. During the daytime, the highest UT mean mixing ratios of isoprene (0.38 ± 0.27 ppbv), its OP (0.26 ± 0.22 ppbv) and monoterpenes (0.053 ± 0.02 ppbv) are observed between 11 to 12 km. Due to active photochemistry during daytime, the mean product/precursor ratio for isoprene-OP/isoprene (1.58 ± 1.09) is almost double that observed at nighttime (0.83 ± 0.62) (Supplementary Fig. 5). At night, approximately 21% and 27% of the

mean boundary layer values of isoprene and monoterpenes remain in the UT. In comparison, only 15% of boundary layer isoprene-OP reached the UT, presumably due to in-cloud losses associated with its higher solubility and possible subsequent enhanced removal through in-cloud aqueous oxidation chemistry[38]. Wang et al.[39] investigated the ozonolysis of methyl vinyl ketone (MVK) and methacrolein (MACR) at the air-water interface, and observed that interfacial water molecules act as a catalyst for the reaction between MVK and $O_3$. Thereby, the primary isoprene oxidation products (MVK and MACR) can potentially undergo aqueous phase chemistry to produce further even more soluble carbonyl compounds, including oxalate, glyoxal, and methylglyoxal[38-40]. Our results support the notion that isoprene-OP is preferentially removed in convection relative to isoprene, and the more soluble second-generation oxidation products were not observed at high altitudes. During the daytime, the monoterpenes reaching the UT constitute 23% of the boundary layer values. These values are comparable to, but higher than, the convectively transported isoprene, which was 17% of its boundary layer value. The loss of monoterpenes during day and night during transport from the boundary layer to UT is, therefore, also lower than that of isoprene. This can be due to the slower reaction rate with OH compared to isoprene and the tendency for the reaction rates of some monoterpenes with $O_3$ to decrease under low-temperature conditions[41,42] in the UT.

From the measurements presented here, it is clear that elevated levels of the surface emitted primary BVOCs (isoprene and monoterpenes) and their photochemical products are present in the Amazonian UT (9-13 km). This is important because the oxidation of these species takes place in the UT, where low temperature (-60 °C), low pressure (200 mbar), and limited particle surface area for condensation create favorable conditions for new particle formation. Curtius et al. (2024)[16] have reported extremely high particle production rates for a convective system, and the NASA Atmospheric Tomography (ATom) Mission have reported unexplained high particle numbers emerging from the Amazon region[35,43,44]. It should be noted that in time, a fraction of these newly formed particles can grow into larger sizes through coagulation and condensation and generate cloud condensation nuclei.

## Diel variations

A remarkable feature of this dataset is that it allows investigation of the diel (24-hour) variation of the three selected species at multiple

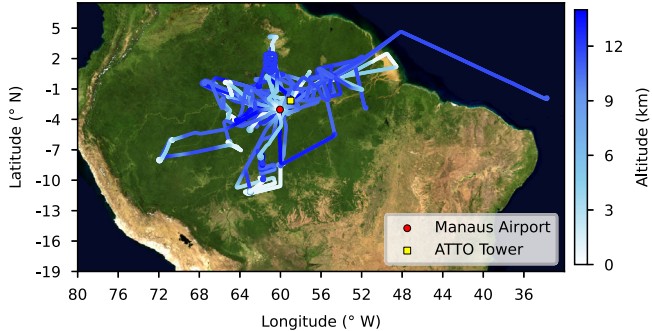

**Fig. 1 | The flight tracks during the CAFE-Brazil campaign over the Amazon rainforest.** The flight track is color-coded by altitude. All flights were performed from Manaus airport (red circle), Brazil (3° S, 60° W). The location of the Amazon Tall Tower Observatory (ATTO) ground-based forest site is marked with a yellow square. Satellite picture data in this figure obtained from https://wvs.earthdata.nasa.gov, NASA Worldview Snapshots.

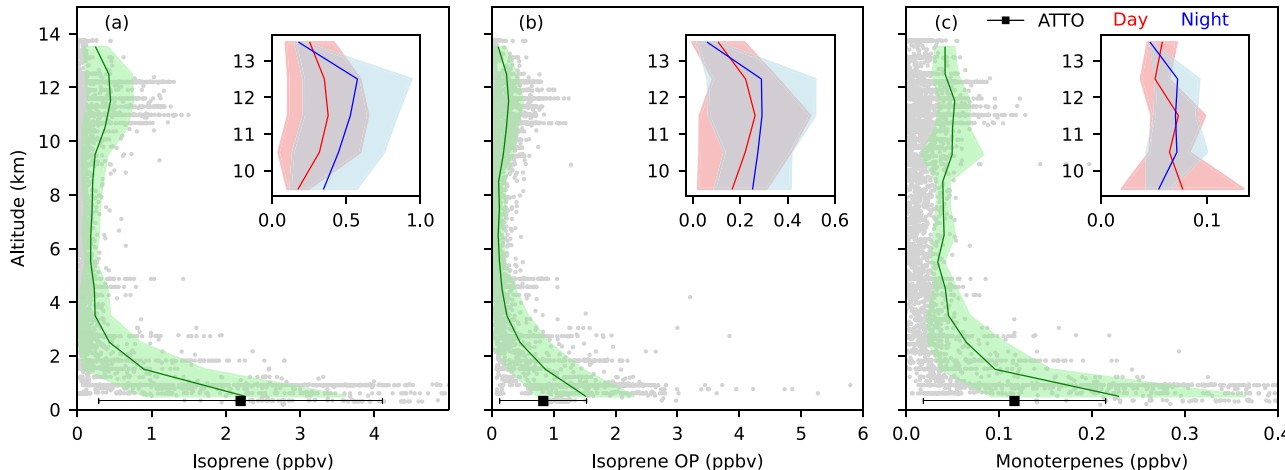

**Fig. 2 | Vertical profiles of biogenic volatile organic compounds (BVOCs) during the CAFE Brazil campaign.** Vertical profiles of **a** isoprene, **b** isoprene oxidation products (OP) and **c** monoterpenes during the CAFE Brazil campaign. Gray dots are 1 min-averaged data, while mean values are represented by a solid green line (1 km bin); the shaded light-green area shows the standard deviation. Note the mean value is calculated only for values above the limit of detection (LOD). The black square with error bars represents the mean value of Amazon Tall Tower Observatory (ATTO) data measured during the flight days. The inset plots show the mean vertical profiles of BVOCs at higher altitudes (9–14 km) during day- and nighttime in red and blue, respectively.

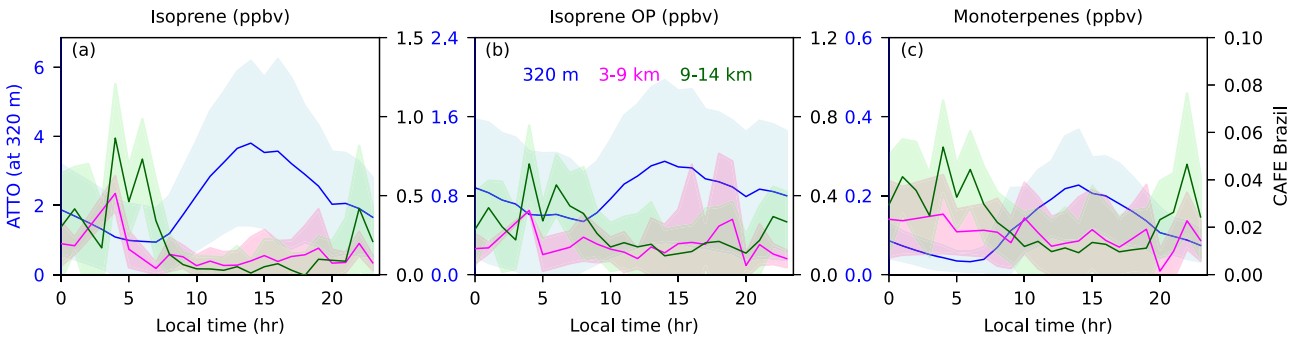

**Fig. 3 | Diel cycle of biogenic volatile organic compounds (BVOCs) for three altitudes over the Amazon rainforests.** Diel cycles of **a** isoprene, **b** isoprene oxidation products (OP) and **c** monoterpenes for three different altitudes regimes of 320 m (left side y-axis) at Amazon Tall Tower Observatory (ATTO), 3–9 km, and 9–14 km (right side y-axis) during the CAFE-Brazil campaign.

altitudes, namely in the Lower Troposphere (LT), middle troposphere, and Upper Troposphere (UT). The diel variations of isoprene, isoprene-OP, and monoterpenes at three altitude regimes of 320 m, 3–9 km, and 9–14 km are shown in Fig. 3 and Supplementary Fig. 6. The ATTO data measured at 320 m during the flight days was used to determine a diel (24-hour) variation at low altitudes. In the lowest altitude regions, all presented BVOCs exhibit strong diurnal variations with peak mixing ratios of isoprene and monoterpenes measured in the afternoon (12:00–16:00), coinciding with the highest solar radiation and temperature. This is consistent with fresh biogenic emissions entering the well-mixed boundary layer, both from photosynthetically driven de-novo emissions and temperature-driven release from storage pools with the plants. Similar temporal patterns were observed for mixing ratios of isoprene-OP, indicating its rapid formation through the photochemical oxidation of isoprene by OH. Estimates of OH radical concentrations at this site indicate that they also peak between 14:00 and 15:00[45]. High secondary production of isoprene-OP at this time causes a slower decline of isoprene-OP than the precursor isoprene. For the measurements at 320 m, the lowest values of BVOCs occur in the early morning (04:00-06:00).

In the middle troposphere, we observe substantially lower values for all three species, with a modest peak in isoprene visible at 04:00, while isoprene-OP shows peaks both at 04:00 and later and with more variability between 15:00 and 19:00 local time. The early morning peaks may be associated with the re-assimilation of the residual layer that is left isolated aloft when the nocturnal boundary layer forms lower down. The levels of isoprene and monoterpenes are very low during the daytime in the middle troposphere. This is consistent with model predictions for the altitude of maximum OH being around 3-4 km[46,47] and, therefore, efficient loss through oxidation during daytime. The same processes are responsible for the late afternoon elevated OP.

Most interestingly, a diel cycle is also observed in the UT (9-14 km), but in contrast with the near-surface values which peaked between 14:00 and 15:00, the high-altitude maxima occur shortly before dawn between 03:00 and 06:00. It is important to note that deep convection events were widespread and frequent throughout the Amazon basin both by day and night (Supplementary Fig. 7). Within such convection, air from the boundary and residual layers may be transported rapidly to higher altitudes on timescales of 0.5–2 hours. However, the atmospheric lifetime of isoprene with respect to OH by day is also 1-2 hours. This means that isoprene (and the other BVOCs) will undergo considerable photochemical loss during the horizontal transport following outflow from the cloud by day. In contrast, at night, any convection occurring from sunset to sunrise can export unreacted residual isoprene (and other BVOCs) to high altitudes without subsequent loss from OH. $O_3$, which is still present by night, reacts much slower with isoprene (lifetime about 2 days at 20 ppbv

$O_3$)[48,49] and boundary layer $O_3$ levels in the Amazon are typically low 10–20 ppbv[50,51].

In the UT, the highest mean mixing ratios of isoprene, its oxidation products and monoterpenes are measured at $0.86 \pm 0.34$, $0.56 \pm 0.20$ and $0.05 \pm 0.02$ ppbv, respectively, at 04:00 local time, two hours before dawn. The mean mixing ratio of monoterpenes in the UT at this time is closely comparable to those measured in the boundary layer at 320 m between 04:00 and 06:00 local time, suggesting that very efficient nighttime vertical transport has occurred. Isoprene transported by convection at night can accumulate in the UT since $O_3$ in the outflow will be low and OH is not present. However, from dawn onwards, the influx of solar energy increases, promoting outflow plume dispersion, in-mixing of $O_3$, and the photochemical formation of OH. The nocturnally transported isoprene and monoterpenes decline rapidly to low mixing ratios after 08:00 local time when OH production increases. In contrast, isoprene OP decreased more slowly due to its secondary production from the isoprene oxidation, partially offsetting loss through reaction with OH. The higher mean mixing ratios of isoprene-OP in the UT result from both the updraft during nighttime and their in-situ formation from isoprene. It is remarkable that not only do the emissions from the Amazon rainforest biome extend to 14 km and influence the UT, but we also observe distinct diel cycles in the BVOCs at different altitudes. The buildup of highly reactive species, such as isoprene and monoterpenes during the night primes the upper atmosphere for vigorous photochemistry at dawn[15].

## BVOCs sensitivity analysis in the Amazonian atmosphere

The EMAC model simulations were performed to investigate the sensitivity of atmospheric chemistry above the Amazonia basin and the radiative effects to changes in BVOC emissions. The motivation of the sensitivity study is the elevated levels of BVOCs measured in the UT associated with convection and addressing how changes in the BVOC emission at the surface will impact the atmospheric chemistry from the boundary layer to the UT (9–14 km). Flight 06 provides notable motivation for this sensitivity study. In this flight, in fact, the measurements documented BVOCs over deforested areas converted to pasture, and contrasted to nearby pristine forests sampled at relatively low altitudes (~900 m) in the boundary layer over Rondônia, Brazil. Figure 4a–d depict the variations in the altitude and mixing ratios of isoprene, isoprene-OP and monoterpenes along flight 6 over forested and deforested regions, respectively.

The flight 06 data show a steep spatial gradient of isoprene mixing ratio over forested and pasture regions. The mean mixing ratios of isoprene, isoprene-OP, and monoterpenes over deforested regions were $0.75 \pm 0.45$, $1.2 \pm 0.36$ and $0.08 \pm 0.04$ ppbv, respectively. In contrast, the mean mixing ratios of isoprene ($2.96 \pm 0.72$ ppbv), and

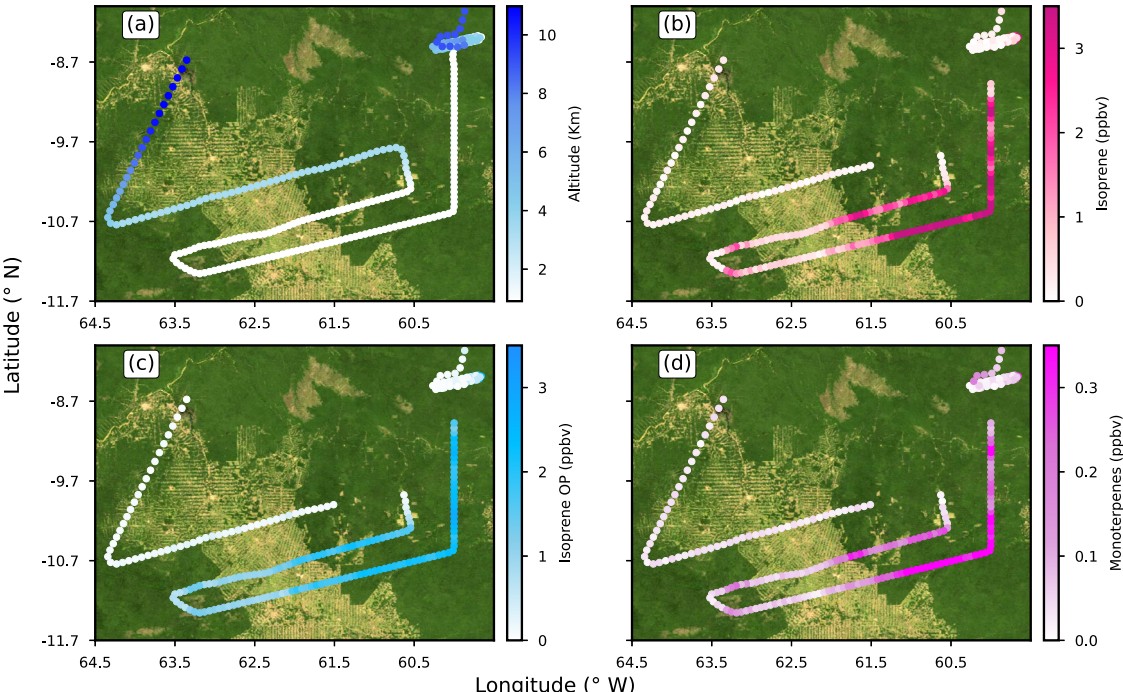

**Fig. 4 | Flight tracks of flight 06 over forested and deforested regions.** Flight tracks of flight 06 with color bar **a** altitude in kilometers and mixing ratios of **b** isoprene **c** isoprene oxidation products (OP) and **d** monoterpenes over the Amazon rainforest. Green and tan colors on the map show the forested and deforested regions, respectively. Satellite picture data in this figure obtained from https://wvs.earthdata.nasa.gov, NASA Worldview Snapshots.

monoterpenes (0.31 ± 0.09 ppbv) over forested regions were approximately four times higher than the pasture regions. While mixing ratio of isoprene-OP (2.12 ± 0.41 ppbv) were only ~1.7 times higher than over the deforested regions. The lower spatial gradients of isoprene-OP are likely due to the transport of the longer-lived isoprene-OP from adjacent forested regions.

Deforestation has been reported as a primary driver of the reduction in BVOC emissions in tropical forests although other factors, such as biomass burning and climate change, also play significant roles. On the other hand, several studies suggested that global BVOC emissions will likely increase with climate change, potentially by up to 50% under scenarios with doubled $CO_2$ concentrations[52,53]. As a result, the future Amazonian atmosphere could experience dramatic positive or negative variations in BVOC emissions. Therefore, we evaluate the atmospheric response in scenarios where Amazonian BVOC emissions are reduced by 50% and 75%, and increased by 50% using the EMAC model.

EMAC is a general circulation model, here used at roughly 200 × 200 km horizontal resolution and 90 vertical levels up to 80 km. BVOCs are emitted using the Model of Emissions of Gases and Aerosols from Nature (MEGAN)[3] and a comprehensive atmospheric chemistry mechanism (Mainz Isoprene Mechanism, MIM)[54] is used. Details of the model are provided in the method section. We performed a base-case scenario and three sensitivity runs with perturbed BVOC emissions, as stated above. The base-case EMAC run generally agreed with the isoprene measured during the CAFE-Brazil campaign, especially in the LT (see Supplementary Fig. 8). While monoterpenes are somewhat underestimated in the boundary layer, most probably due to their reactions with $O_3$, which is known to be overestimated in EMAC, especially in the tropics[55], as well as in other global chemistry-climate models[56]. The measurements show higher BVOC levels in the UT than the model, which can be expected since the aircraft actively sought convective outflow. These sub-grid scale processes are parametrized in the model and averaged over the horizontal extent of the grid box.

However, surface monoterpene and isoprene mixing ratios were evaluated with measurements from the ATTO tower, and simulated ozone, OH and organic aerosol were additionally evaluated with background aircraft observations, showing good agreement (supplementary Figs. 9–12). Moreover, previous studies have evaluated the EMAC model with similar setups for aerosols[57-60], ozone and OH[55]. The simulated diurnal cycle of isoprene mixing ratios (Supplementary Fig. 13) supports our findings observational analysis. The highest isoprene mixing ratios in the BL are simulated in the afternoon hours, with delayed transport to the residual layer, and the highest UT isoprene mixing ratios in the early morning, following night-time convection. Isoprene mixing ratios react non-linearly to changes in BVOC emissions, with a positive feedback due to the increased (decreased) OH availability, the main sink of isoprene, following decreases (increases) in BVOC emissions (Fig. 5)

The annual and seasonal changes in the Amazonian region over 0–20 km altitude, relative to the base case, for the oxidants OH radicals, $O_3$, and the global radiative effect due to aerosol, are illustrated in Fig. 5 and Supplementary Fig. 14, respectively. Each panel in Fig. 5 indicates the relative change across the atmospheric column. The results show that BVOCs strongly influence the oxidation capacity of the atmosphere, even up into the UT. Annually, the strongest effects on OH (changes of 300% relative to the base case, when BVOC emissions were lowered to 25%), occurred in the boundary layer at 0–3 km altitude. The highest relative change in OH (~400%) in the LT (0–3 km) during the dry season suggests a higher loss rate of OH, driven by elevated BVOC emissions over the Amazon basin during this period (Supplementary Fig. 14). In the mid-troposphere (5–10 km), a 75% decrease in BVOC emissions leads to a 100-150% relative increase in OH mixing ratios. On the other hand, OH only decreases by 15–40% after increasing BVOC emissions by 50% at these altitudes. This relative change in OH after reducing and increasing BVOC emission shows a non-linear dependency (Fig. 5b). An increase in BVOC emissions can lead to a reduction in OH levels; however, the sensitivity of OH to

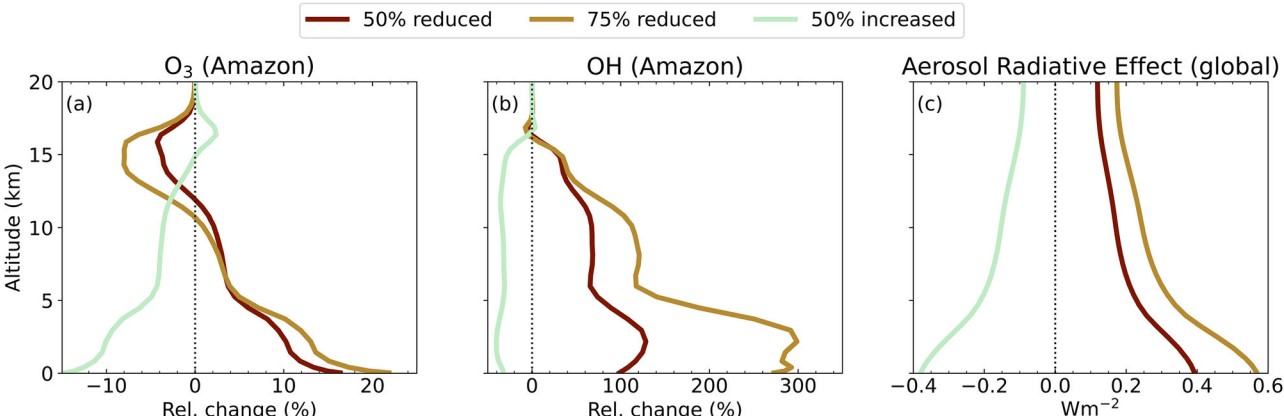

**Fig. 5 | The relative change in the vertical profiles of ozone, OH radical and aerosol radiative effect due to change in the biogenic volatile organic compounds (BVOCs) emission at the surface.** The vertical profiles of relative changes in **a** $O_3$ and **b** OH over the Amazon region and of absolute changes in **c** the global radiative effect due to aerosol between the base simulation and the simulation with 50% and 75% reduction and a 50% increase in the BVOCs emission. The radiative effect is the imbalance between incoming and outgoing radiation and is calculated at each vertical level.

increases in BVOC is buffered, as other sinks and competing processes also play a significant role[47]. It should be noted that such significant increases of OH in this region will also affect the lifetimes of longer-lived radiatively active gases such as methane and reactive halo-carbons. Indeed, the model estimates that the methane lifetime will decrease by ~0.3–0.4 years globally if forest BVOC emissions over the Amazon are reduced by 75%. Scott et al.[61] estimated a 0.5-year reduction in methane lifetime due to global deforestation, in close agreement with these results.

Interestingly, the net annual changes in $O_3$ vary strongly with altitude. When Amazonian BVOC emissions are reduced by 75% in the model, $O_3$ increases by circa 22% in the LT (0–4 km) averaged over the Amazon basin (Fig. 5a), with local increases exceeding 30% (Supplementary Fig. 15), but decreases by 10% in the UT (between 13 and 16 km). Seasonally, the strongest influence on $O_3$ at the surface is observed during the dry season (>25% relative change), while this impact is more pronounced (~ -10%) in the UT during the wet season and the transition from dry to wet periods (Dec–Jan, Supplementary Fig. 14). This indicates that the significant convective transport of BVOCs during the wet seasons plays a pivotal role in the production of $O_3$ in the UT. The $O_3$ increase in the lower troposphere can be attributed to lower rates of direct BVOC-$O_3$ reactions, particularly for monoterpenes that react faster with $O_3$ than isoprene[62]. Therefore, we infer that BVOCs act mainly as the sink for $O_3$ (Supplementary Fig. 15) because the rainforest boundary layer is a NOx poor environment, where photochemical $O_3$ production is limited by extremely low NO[50,51]. In contrast, the high-altitude regions are enriched in NOx by lightning, and the reduction in $O_3$ in the UT due to the reduction in BVOC levels reveals that $O_3$ production is limited by BVOCs, and that lower surface emissions mean less BVOC is available aloft to drive photochemical $O_3$ production.

We also estimate the radiative effect of perturbations in BVOC emissions over the Amazon region specifically due to the changes in $O_3$ and aerosol. The changes in radiative forcing due to aerosol are also shown in Fig. 5c as a function of altitude. It is remarkable that a 75% BVOC emission reduction on the Amazon leads to a very strong regional warming, with aerosol radiative effect of ~6 Wm$^{-2}$ at the surface over the Amazon Basin and a lower but significant ~ 0.6 W m$^{-2}$ globally. In contrast, a 50% increase in BVOC emissions has a substantial global cooling effect of -0.4 W m$^{-2}$ (Fig. 5c). By performing additional sensitivity studies reducing only isoprene and only monoterpenes (supplementary Fig. 16), we learn that isoprene dominates the observed changes in the gas phase oxidants. In contrast, the

monoterpenes play a main role in the aerosol radiative forcing. By reducing regional BVOCs up to 75%, a small radiative cooling occurs in the global upper troposphere (-0.03 W m$^{-2}$), due to ozone, which is relatively insignificant compared to the changes caused by aerosols.

Previous modelling studies have reported a latitudinal sensitivity to deforestation, with tropical deforestation accounting for 80% of the direct radiative forcing. Scott et al.[61] have also reported that BVOCs can impact short-lived tracers, especially aerosol, $O_3$ and methane, noting the negative radiative forcing resulting from the reduction in $O_3$ and methane concentrations while noting positive radiative forcing attributable to low aerosols[63]. The aforementioned modelling results were conducted with a different model and methodical approaches, but nevertheless, the general results are in good agreement with those shown here. The results of BVOC sensitivity study emphasize the importance of the Amazon rainforest for the global climate system.

This study highlights the role of convection in shaping the atmospheric distribution of BVOCs and their contribution to upper tropospheric chemistry, including $O_3$ production, and OH loss. However, the radiative warming that occurs when monoterpene-generated aerosol is reduced overwhelms the small (5%) changes from the methane lifetime change. It should also be noted that the Amazon rainforest stores approximately 108 Pg C in living aboveground biomass[25,64]. According to this study, if deforestation through burning accounts for the 50% reduction in Amazonian biomass, it would additionally release approximately 54 Pg C (equivalent to a global increase of 25 ppm $CO_2$) into the atmosphere, leading to a strong positive radiative forcing and warming, as well as a long-term perturbation of the global hydrological and carbon cycles. The potential effects of deforestation and afforestation on the Earth's atmosphere are further discussed elsewhere, e.g., in Weber et al. (2024)[65]. Our BVOC sensitivity study highlights the role these species play in the atmosphere. In short, by storing carbon and releasing BVOCs, the Amazon rainforest modulates atmospheric $CO_2$ concentrations and regulates oxidation processes in the upper and lower troposphere.

This study describes a unique and comprehensive dataset for isoprene, its oxidation products (isoprene-OP), and total mono-terpenes over the Amazon rainforest from 300 m to 14 km altitude during a two-month airborne and ground-based field campaign (CAFE-BRAZIL, December 2022–January 2023). The data reveal that tropical deep convection has a strong impact on the distribution of these short-lived trace gases, with little influence in the mid-troposphere (3–9 km) but elevated levels in the upper troposphere (UT, 9–14 km). Moreover, we show that nocturnal convection effectively transports biogenic

species to the UT, where they accumulate through the night without effective oxidation, and then rapidly oxidize at daybreak. The results support the conclusions of Palmer et al., who used a combination of satellite and model data to infer nocturnal vertical transport of isoprene (6–8 km), although our in-situ data shows vertical transport and outflow through convection occurring at significantly higher altitudes (10–12 km). These results also show that the UT isoprene values highlighted by Curtius et al.[16] in a dawn-flight convective outflow case study with associated strong particle production are widespread over the region.

By comparing surface and UT levels of isoprene, isoprene-OP and monoterpenes, we note that convection preferentially removes the more oxidized and soluble isoprene-OP, relative to the primary emissions. Here we provide the vertical and diel (24-hour) resolved distribution of these species, showing that the daily peak in the concentration of isoprene shifts from 12:00 to 14:00 at the surface to around 04:00 at 10 km, again due to deep nocturnal convection and absence of photochemical degradation. By selecting realistic values based on measured BVOC mixing ratios over deforested regions during this campaign and previously reported projections, the sensitivity study highlights that a reduction in BVOC emissions over the Amazonian region would lead to a decrease in upper tropospheric $O_3$, an increase of lower tropospheric OH, a shortening of the methane lifetime, and a strong net increase in radiative forcing through $O_3$ and aerosols. An underestimation of the model results compared to the observations suggests that the effects of BVOCs on $O_3$, OH, and radiative forcing in the UT may be stronger than the relative changes presented in this study. This highlights the need to include BVOC chemistry in global models when simulating global change and their potential contributions to radiative forcing.

## Methods
### Instrumentation
The high time- and mass-resolved BVOC measurements of isoprene, isoprene oxidation products (OP) and total monoterpenes were performed using PTR-TOF-MS (Ionicon 8000, Analytik GmbH, Austria). In the aircraft, a 2 m long 1/8-inch sulfinert inlet tube was used for sampling and was permanently heated to 40 °C. Hydronium ions ($H_3O^+$) were used as a reagent ion to ionize molecules with sufficient proton affinity in the ambient air molecule within the PTR-drift tube. The instrument is operated with a drift pressure of 2.2 mbar, E/N 137 Td, drift voltage of 600 V, and reactor temperature of 60 °C, running at time resolution of 1 s. An air molecule (R) is ionized in the drift tube by the following proton transfer equation

$$H_3O^+ + RH \rightarrow RH^+ + H_2O$$

Molecules with proton affinities higher than that of water (693 kJ mol⁻¹), can be ionized in the drift tube and therefore measured by the detector. In flight background measurements were conducted for 10 min at every 1–2 km of altitude using pressurized synthetic air in the aircraft. The ionicon data analyser (IDA) 2.2.0.4 software was used to analyze the raw data. The instrument was calibrated the day before each flight. Isoprene, Isoprene-OP and monoterpenes were calibrated with a commercial gravimetrically prepared standard (Apel-Riemer Environmental Inc) containing isoprene, methacrolein, $\alpha$-pinene and other compounds. The calibration was performed for both dry and humid air conditions and humidity dependencies were factored into the final data. The limit of detection (LOD, 3σ) of averaged 1 min data for isoprene, isoprene OP (MVK, MACR and ISOPOOH) and monoterpenes ranges from 85 to 180 ppt, 13 to 56 ppt, and 24 to 43 ppt for different flights, respectively. Data below the detection limit were filtered out during the averaging. The total uncertainty, including both precision and calibration error, was estimated to be approximately

10-20%. Several studies have shown that isoprene detection by PTR-TOF-MS can be compromised by interferences caused by the fragmentation of larger molecules or ozone generated species. Therefore, we used data measured by the fast Gas chromatography-mass spectrometer (GC-MS) onboard to validate isoprene values and to assess the degree of interference in ambient air over the Amazon. PTR-TOF-MS isoprene validation with the GC-MS and associated correction are discussed in detail in supplementary text 1. The detailed information of GC-MS is discussed in Supplementary Text 2.

### Model
Model calculations were performed using the ECHAM/MESSy Atmospheric Chemistry-climate Model (EMAC)[54,66]. Simulations were conducted at a horizontal resolution of T63, equivalent to 1.875° x 1.875° (approx. 200 × 200 km) with 90 hybrid, terrain-following vertical levels extending up to 0.01 hPa (~80 km). Simulations were nudged towards meteorological reanalysis data (ERA5)[67] to represent actual meteorological conditions.

Gas-phase chemistry was simulated using the MECCA[68] submodel with MIM1[54,69] chemistry. Photochemical J-values were calculated using the JVAL[70] submodel. Lightning NOx emissions were computed using Grewe's algorithm[71], while dry deposition, wet deposition, and sedimentation were modeled with the DDEP[72], SCAV[73], and SEDI[72] submodels, respectively.

Anthropogenic surface emissions of reactive gases and primary aerosols were sourced from the CEDS database[74] for the year 2019, supplemented by aircraft emissions from the CAMS global aviation dataset[75]. Natural emissions of biogenic VOCs were calculated using the MEGAN submodel[3]. For the sensitivity studies, BVOC emissions were scaled down to simulate the impact of deforestation. For the sensitivity studies, BVOC emissions were simply scaled with a factor of 0.5 (50% reduction), 0.25 (75% reduction), and 1.5 (50% increase) without further changes to the simulation.

The model's aerosol microphysics rely on seven interactive lognormal modes, comprising four hydrophilic and three hydrophobic modes, to represent the aerosol size distribution. All aerosols are treated as spherical particles[76]. While aerosols are externally mixed across size modes, they are internally mixed within each mode based on contributing species. Gas-aerosol partitioning is determined using ISORROPIA for inorganic species[77] and ORACLE for organic species, based on their sources and volatility[78]. Radiative calculations are performed with the RAD submodel[79], comparing the sensitivity to the reference simulation, with aerosol optical properties calculated with the AEROPT submodel[79].

## Data availability
All data that support the findings of this study are publicly available on FigShare (https://doi.org/10.6084/m9.figshare.28692260.v1)[80]. Source data are provided with this paper.

## Code availability
The Modular Earth Submodel System, (MESSy, https://zenodo.org/doi/10.5281/zenodo.8360186)[81] is continuously further developed and applied by a consortium of institutions. The usage of MESSy and access to the source code is licenced to all affiliates of institutions which are members of the MESSy Consortium. Institutions can become a member of the MESSy Consortium by signing the MESSy Memorandum of Understanding. More information can be found on the MESSy Consortium Website (http://www.messy-interface.org).

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

## Acknowledgements

The authors acknowledge the support by the German Aerospace Center Flight Experiments facility (DLR-FX) for operating HALO, specifically the

pilots, engineers, technicians, and the operations team. The funding for this research has been received from the Max Planck Society and the German Research Foundation (DFG) for the CAFE-Brazil campaign. The ATTO tower research has been supported by the Max Planck Society, the Bundesministerium für Bildung und Forschung (BMBF contracts 01LB1001A, 01LK1602B, and 01LK2101B), the Brazilian Ministério da Ciência, Tecnologia e Inovação (MCTI/FINEP contract 01.11.01248.00), the Conselho Nacional de Desenvolvimento Científico e Tecnológico (CNPq, Brazil) (process 200723/2015-4). We acknowledge the use of imagery from the Worldview Snapshots application (https://wvs.earthdata.nasa.gov/), part of the Earth Observing System Data and Information System (EOSDIS). N.T. acknowledges funding from the Alexander Von Humboldt Foundation. Special thanks to Thomas Klüpfel and Giovanni Pugliese for assisting with VOC measurements during the CAFE-Brazil campaign, and to Joseph Byron for his contributions to the lab experiment for isoprene correction. We would also like to especially thank all the people involved in the technical, logistical, and scientific support of the ATTO project.

## Author contributions

N.T. analyzed the VOCs data, with contributions from B.E.K., A.E., N.W. and A.R. N.T. wrote the original draft. M.K. led the modeling work, including simulations and writing, with support from R.V. and A.P. J.W. supervised the study and interpretation. J.L. and L.A.T.M. supervised the research that led to this study. All authors contributed to editing the article and approved the submitted version.

## Funding

## Competing interests

The authors declare no competing interests.
