## [Transparent Peer Review file · Nature Communications]

Impacts of convection, chemistry, and forest clearing on biogenic volatile organic compounds over the Amazon

Corresponding Author: Dr Nidhi Tripathi

Version 0:

Reviewer comments:

Reviewer #1

(Remarks to the Author)

General comment:

The manuscript is mainly based on discussions of BVOC and ozone measurements from the CAFE-Brazil experiment during December 2022–January 2023. Measurements in the Amazon rainforest have become even more important considering the urgent scenario we find ourselves in regarding reducing and predicting climate change and the extreme events resulting from that. The paper has a very good description of the measurements and a valid scientific question to be further investigated. My main concern is related ensuring the scientific contribution of the paper is well supported. The link-approach and logic sequence in developing and discussing the papers conclusion are not clear and well aligned. There are some technical and scientific issues that need to be addressed for the authors before I can give my full recommendation for publishing the manuscript.

Specific comments/questions:

- I recommend the authors work to define what it the main goals/contributions they want from this paper:

(i) deep convection events transporting BVOCs to the troposphere and how they stay there due to residual boundary layer and low oxidant variabilities, so in the early morning those contributions trigger NPF formation. The authors can show this using the CAFE-Brazil data and using the model to investigate how much this “base scenario” would change if the BVOCs have reduced emissions. To do that the authors need to first show the model is minimally capable of reproducing background conditions for gases and particles. Once the model can reproduce BVOCs concentrations (evaluate process with observational data), then the sensitive modeling experiment can be performed.

(ii) Only focusing on the observational side for BVOCs, O₃ and NO_x. In this case, I strongly recommend the authors add to their analysis the set of important data from the GoAmazon 2014/2015 experiment and the data from ATTO tower since 2013 (ozone, NO_x and VOCs). This would give more robust statistics to evaluate future consequences for a scenario with BVOC reduction, it's important to analyze how those compounds change seasonally as well. It's not clear that the conclusions reached in this paper are true generally, or just a result of some seasonal or annual variation in the two months used by the authors. Additional observations would alleviate these concerns.

- How did the authors define the convection events (Convective transport of air from low altitudes)?

- The authors should discriminate better between urban and forest areas during the flight pathway and explore more the differences in the measurements in these areas. The area where the CAFE-Brazil experiment was developed has a city (almost 12km²) with more the 2 million people and it has a huge impact on the photochemistry and oxidation capacity around the forest. The 2 following papers about the dynamics of O₃ production in the Amazon rain forest:

Shilling, J. E.; Pekour, M. S.; Fortner, E. C.; Artaxo, P.; de Sá, S.; Hubbe, J. M.; Longo, K. M.; Machado, L. A. T.; Martin, S. T.; Springston, S. R.; Tomlinson, J.; Wang, J. Aircraft observations of the chemical composition and aging of aerosol in the Manaus urban plume during GoAmazon 2014/5. *Atmos. Chem. Phys.* 2018, 18,10773–10797.

Nascimento, J. P.; Barbosa, H. M.; Banducci, A. L.; Rizzo, L. V.; Vara-Vela, A. L.; Meller, B. B.; Gomes, H.; Cezar, A.; Franco,

M. A.; Ponczek, M.; Wolff, S.; Bela, M.; Artaxo, P. Major Regional-Scale Production of O₃ and Secondary Organic Aerosol in Remote Amazon Regions from the Dynamics and Photochemistry of Urban and Forest Emissions. *Environ. Sci. Technol.* 2022, 56, 9924– 9935, DOI: 10.1021/acs.est.2c01358

- The authors should show evidence of the claimed deep outflow events and associate them with the BVOCs (before, during and after the event).

- The follow paragraph has a reference that seems unrelated to its content “To assess the full atmospheric effects of deforestation, the chemical and physical impacts of BVOC must be characterized and understood”

- Line 98, the authors should define what “MR” is.

- Figure 02: The image needs to be improved. The resolution is very poor making it hard to read the inset plots. If possible, vector images would be much better. The inset plots right plot with the day (red)/ night (blue) are confusing to me because the y axis doesn't go to lower than 9km and BVOC values at the PBL high are bigger in the morning due to the emissions being correlated with solar radiation. The authors should add metadata that describes how the plots from the flight were made: eg include range, hours, location, days, etc

- Figure 03: Replacing this with a vector image would be very helpful in order to easily compare the difference in the measurements between the heights. Additionally, the region the measurements are taken needs to be specified. Is it urban or forest? If the airplane is moving how can data be compared with a stationary source?

- Lines 206-207 “It is important to note that deep convection events were widespread and frequent throughout the Amazon basin both by day and night.”. Can the authors show data evidencing this premise? It would enrich the paper discussion and scientific contribution for the community, especially due the fact all these events are yet not well defined and understood. A field campaign such as CAFE-Brazil is a great opportunity to help on better understanding those processes.

- Figure 04: This figure needs to be rethought because is very confusing, the color scale of the 2 different flights is not the same. The brown-green for deforested and forest areas are not very obvious (maybe call it green/tan?). The figures should show a bigger picture of the area where the flights took place the urban/forest/deforested area could be more highlighted. It might be easier to understand this data and support the authors point if it was a scatter plot showing altitude vs atmospheric constituent (isoprene, monoterpene, etc) and the points were colored green or brown for forested/deforested.

- Line 295-297 “ Interestingly, the net changes in O₃ vary strongly with altitude. When Amazonian BVOC emissions are reduced by 75% in the model, O₃ increases by circa 10% in the LT (0-4 km) but decreases by about the same amount in the UT (between 13-16 km).”

The authors should clarify in which location the O₃ shows the increase (10%) due to the BVOCs reduction. In forest regions such as ATTO tower region, the O₃ production is limited by NO_x emissions, so a reduction on BVOCs is unlikely to be the cause of the O₃ increase. Variations in NO_x are more aligned to changes in O₃ concentration on NO_x-limited regions. I recommend the authors:

(i) Distinguish the regions when the O₃ was increased

(ii) Be more specific with the time range related to the increase.

(iii) Compare ozone observational data with simulations (without the BVOCs reduction), so the analyses would be more realistic when comparing ozone concentrations.

- The authors should consider the finds in the 2 following papers about the dynamics of O₃ production in the Amazon rain forest:

Shilling, J. E.; Pekour, M. S.; Fortner, E. C.; Artaxo, P.; de Sá, S.; Hubbe, J. M.; Longo, K. M.; Machado, L. A. T.; Martin, S. T.; Springston, S. R.; Tomlinson, J.; Wang, J. Aircraft observations of the chemical composition and aging of aerosol in the Manaus urban plume during GoAmazon 2014/5. *Atmos. Chem. Phys.* 2018, 18, 10773–10797.

Nascimento, J. P.; Barbosa, H. M.; Banducci, A. L.; Rizzo, L. V.; Vara-Vela, A. L.; Meller, B. B.; Gomes, H.; Cezar, A.; Franco, M. A.; Ponczek, M. Major Regional-Scale Production of O₃ and Secondary Organic Aerosol in Remote Amazon Regions from the Dynamics and Photochemistry of Urban and Forest Emissions. *Environ. Sci. Technol.* 2022, 56, 9924– 9935, DOI: 10.1021/acs.est.2c01358

- Line 297-300 “The O₃ increase close to the surface can be attributed to lower rates of direct BVOC- O₃ reactions, particularly for monoterpenes that react faster with O₃ than isoprene. The rainforest boundary layer is a low NO_x environment, where photochemical O₃ production is limited by extremely low NO_x.”

I suggest the authors make numerical experiments changing the NO_x availability in the forested regions. The ozone variation here is moderated by the NO_x concentration not BVOCs.

- Line 309-311 “We also estimate the radiative forcing changes over the Amazon region (calculated at the ATTO location) associated with the reduction of Amazon BVOC emissions by 50 and 75%, specifically due to the changes in O₃ and aerosol”

The authors estimated the radiative forcing based in only one data point (ATTO tower location) in the Amazon region, which does not represent the whole Amazon region. I recommend the authors to make this clearer.

- Have the authors compared O₃ and aerosol observational data in the ATTO region with modeling result, before analyzing the changes due to the BVOCs emissions reductions (50% and 75%)? To claim that the radiative forcing changes over the Amazon region (because the reduction of Amazon BVOC emissions) are due changes in O₃ and aerosol its necessary to ensure the control model results agree with aircraft measurements. Shrivastava (2019) and Nascimento (2022) compare model and aircraft measurements by sampling the model data at the aircraft's time and position to make a direct comparison. See the reference below:

Shrivastava, M.; Andreae, M. O.; Artaxo, P.; Barbosa, H. M. J.; Et al Urban pollution greatly enhances formation of natural aerosols over the Amazon rainforest. *Nat. Commun.* 2019, 10, 1046.

- I recommend the authors to consider the difference in VOC concentrations in the Amazon region depending on the season. Concentrations of isoprene (m/z 69) and its oxidation products (m/z 71), biogenic precursors of OA, are a factor of 2–3 higher in the dry season than the wet season. See the following two references:

Gu, D., Guenther, A. B., Shilling, J. E., Yu, H., Huang, M., Zhao, C., Yang, Q., Martin, S. T., Artaxo, P., Kim, S., Seco, R., Stavrou, T., Longo, K. M., Tóta, J., de Souza, R. A. F., Vega, O., Liu, Y., Shrivastava, M., Alves, E. G., Santos, F. C., Leng, G., and Hu, Z.: Airborne observations reveal elevational gradient in tropical forest isoprene emissions, *Nat. Commun.*, 8, 1–7, <https://doi.org/10.1038/ncomms15541>, 2017.

Other references recommended to the authors:

Yáñez-Serrano, A. M.; Bourtsoukidis, E.; Alves, E. G.; Bauwens, M.; Stavrou, T.; Llusià, J.; Filella, I.; Guenther, A.; Williams, J.; Artaxo, P.; Sindelarova, K.; Doubalova, J.; Kesselmeier, J.; Peñuelas, J. Amazonian biogenic volatile organic compounds under global change. *Global Change Biol.* 2020, 26, 4722–4751.

Yáñez-Serrano, A. M. et al. Diel and seasonal changes of biogenic volatile organic compounds within and above an Amazonian rainforest. *Atmos. Chem. Phys.* 15, 3359-3378, doi:10.5194/acp-15-3359-2015 (2015).

Reviewer #2

(Remarks to the Author)

The authors report surface and aircraft measurements of biogenic volatile organic compounds and their oxidation products, collected over the Amazon forest. They also use a global 3-D atmospheric chemistry model to interpret these data and to explore the sensitivity of their results to future deforestation and the corresponding changes in radiative forcing. This reviewer enjoyed the paper and found it scientifically interesting, but found some of the later modelling experiments to be an unnecessary extension of the observational results. I have mostly described very minor comments.

The manuscript is well written but the authors should take pity on this reviewer's eyes and make the figures bigger. Figures 2-4 are too small. The inset figure in Figure 2 is virtually unreadable – given the space in the main figure the author could afford to make the inset figure bigger.

Line 131: How did the authors estimate 20% of isoprene escapes the BL?

Line 139: Fussy, I know, but I do wonder whether “innate” is needed.

Line 171: The NASA Atmospheric Tomography (ATom) Mission should be introduced. Not ATOM.

Line 192: If you're using the 24-hour clock, please use 06:00 not 6:00. Here and elsewhere in the manuscript.

Line 220: Awkward sentence using parentheses.

Line 230: Is it that remarkable? If you built a Hovmoller plot, would you see this delay is associated with the vertical transport of the material? Does ECHAM/MESSy reproduce this result?

Line 246: Can the author back this statement up with a model calculation?

Line 277: This reviewer would argue that this model error is associated more with sub-grid scale vertical motion rather than emissions averaged over large areas. Otherwise you wouldn't have such good agreement in the BL and LT. Worth some additional discussion.

The authors then launch into a series of sensitivity numerical experiments, all of which are interesting although this reviewer suggests they could be more aligned with the main points of the paper associated with the data. At the very least, the authors would need to describe the elevated levels of NO_x associated with biomass burning associated with deforestation, etc.

Line 332: The authors felt obliged to provide a caveat about any inadvertent recommendation to the reader about cutting down the Amazon rainforest, and provided some of the competing impacts associated with such an idea. This reviewer suggests that if they keep this paragraph they should describe all the consequences not just carbon, but also water, albedo, etc. These are covered in more detail in Weber et al: <https://www.science.org/doi/10.1126/science.adg6196>.

Reviewer #3

(Remarks to the Author)

Summary/ General comments:

The paper starts by discussing CAFE-Brazil measurement flights conducted between December 2022-January 2023 over the Amazonian rain forest. They display PTR-TOF measurements of isoprene, monoterpenes and some isoprene oxidation products over a large scale of different altitudes. The vertical profiles presented in this paper shows an enrichment of all three groups in the upper troposphere and that this enrichment happens specifically during the night due to reduced photochemical sink. Hereafter, the focus shifts to two flights performed over regions, where deforestation has taken place previously. The authors discuss two individual flights performed within the boundary layer and describe that monoterpene, isoprene and isoprene oxidation products have lower mixing ratios there as compared to the forested regions. The authors hereafter say that this will also reduce the availability of BVOCs to be transported to the upper troposphere. This leads to the final part of the paper, where O₃, OH and aerosol radiative forcing sensitivities to decreasing BVOC emissions is studied using the EMAC global atmospheric chemistry model, where the BVOC emissions have been either decreased by 50% or 75% over the Amazon or 50% globally. The results show that reducing BVOC emissions influences the oxidant budget in different ways depending on the altitude and the oxidant. Isoprene emission reductions play a key role in influencing the oxidant budget, further decreasing methane lifetime. Decreased emissions of monoterpenes reduce the aerosol radiative effect causing a positive radiative forcing.

Overall, this study comes across to me as a combination of two studies that do not link together perfectly. The deforestation part of the work is interesting, but it would have been better if the authors had addressed also the likelihood of deep convection even taking place over the deforested areas (see e.g., Wang et al., 2009), or shown data on the upper troposphere in the vicinity of these regions. Further on in the modeling part of the work, it would have been interesting if the convective transport of isoprene, isoprene OP and monoterpenes was accounted for in the model in a more sophisticated way, or the effect of deforestation on the likelihood of convection and vertical transport were examined. This would have brought more new insights into the work, as now the rather simple sensitivity study does not necessarily add much to Scott et al. (2018), who did a more comprehensive deforestation study (my opinion).

The paper is interesting and timely due to the high interest of the scientific community on UT NPF and convective transport of isoprene. While not discussed in the paper, the observed elevated mixing ratios of isoprene and some oxidation products are in line with previous research in areas with convective activity (e.g., previous flight campaigns in Amazon and South-Eastern United States) as airborne observations using the PTR-TOF are not unique to this study. Nonetheless, these findings have not been discussed much in the past. What is really remarkable in this work, is that sufficient data exists to derive diel cycles for the three groups that highlight the importance of nocturnal transport of the species to the upper troposphere due to less efficient photochemical sink.

Comments:

1. The Supplementary Fig. 3 shows the observed and simulated isoprene and monoterpene concentrations. You state on L273–274 that they generally agree. Does this mean that this model would not need improvement in terms of the convective transport of isoprene, isoprene oxidation products or monoterpenes? I would personally not say that these agree well (if you plot with x-axis in linear scale). Would be interesting to see how much this difference influences your OH, O₃ budgets or aerosol radiative effect.

2. How did you identify deep convection? Could you show some evidence that the observed elevated UT mixing ratios are associated with it? Could you also specify which observations you refer to on L129-131? Is it isoprene?

3. Many previously published studies could be highlighted:

a. The results from Fig. 2 for isoprene are in line with previously published data from GoAmazon (Shilling et al., 2018; Jo et al., 2021) and SEARC4s from south-eastern US (Jo et al., 2021), which also experience convective activity. There are surely many more.

b. L170–L172: The high particle number concentrations in amazon outflow in the UT, associated with a convective system and isoprene oxidation products, were shown by Zha et al. (2024). The Curtius (2024) paper is not provided within this review file and it is not published, so I cannot evaluate this sentence.

4. Fig. 2 shows the 1 min-averaged data and the 1km bin means. Wouldn't it make more sense to show a median and the interquartile range? The green lines do not really seem representative of the 1 min data. How are the day and night plots made? Could you also show the 1 min data for both and make two panels?

5. It is unclear how the BVOC emission reductions are done for the model simulations. I'm confused by L280-L281— is this just unnecessary repetition? Could you provide a more comprehensive description of the model and the simulations? I think you should also be careful with direct comparisons to Scott et al (2018) or state that what you are studying is not the impact of deforestation.

6. Isn't aerosol radiative effect more accurate term for this kind of study than aerosol radiative forcing?

7. How is the global aerosol RF calculated? How can you have a vertically resolved $W m^{-2}$ when RF should be for the full column? Did you "slice" this somehow? It is also confusing in the text when you talk about "global radiative forcing" if you mean "global aerosol radiative forcing".

Minor comments:

8. Some figures don't have labels, perhaps add the missing?

9. Figure 3: You could also specify that 320m refers to ATTO in the legend for clarity. In the legend, it is hard to distinguish green from blue on my screen at least (plot lines are clear though). Consider increasing linewidth for the legend.

10. L105: chemical oxidation sounds a bit odd, maybe change to chemical loss (oxidation)?

11. Citation 11 seems to be wrong. Perhaps you refer to Williams et al. (2000): An Atmospheric Chemistry Interpretation of Mass Scans Obtained from a Proton Transfer Mass Spectrometer Flown over the Tropical Rainforest of Surinam? The current cited article seems not to include any data on BVOCs.

References

Jo, D. S., Hodzic, A., Emmons, L. K., Tilmes, S., Schwantes, R. H., Mills, M. J., Campuzano-Jost, P., Hu, W., Zaveri, R. A., Easter, R. C., Singh, B., Lu, Z., Schulz, C., Schneider, J., Shilling, J. E., Wisthaler, A., and Jimenez, J. L.: Future changes in isoprene-epoxydiol-derived secondary organic aerosol (IEPOX SOA) under the Shared Socioeconomic Pathways: the importance of physicochemical dependency, *Atmos. Chem. Phys.*, 21, 3395–3425, <https://doi.org/10.5194/acp-21-3395-2021>, 2021.

Scott, C. E., Monks, S. A., Spracklen, D. V., Arnold, S. R., Forster, P. M., Rap, A., Äijälä, M., Artaxo, P., Carslaw, K. S., Chipperfield, M. P., Ehn, M., Gilardoni, S., Heikkinen, L., Kulmala, M., Petäjä, T., Reddington, C. L. S., Rizzo, L. V., Swietlicki, E., Vignati, E., and Wilson, C.: Impact on short-lived climate forcers increases projected warming due to deforestation, *Nat Commun*, 9, 157, <https://doi.org/10.1038/s41467-017-02412-4>, 2018.

Shilling, J. E., Pekour, M. S., Fortner, E. C., Artaxo, P., De Sá, S., Hubbe, J. M., Longo, K. M., Machado, L. A. T., Martin, S. T., Springston, S. R., Tomlinson, J., and Wang, J.: Aircraft observations of the chemical composition and aging of aerosol in the Manaus urban plume during GoAmazon 2014/5, *Atmos. Chem. Phys.*, 18, 10773–10797, <https://doi.org/10.5194/acp-18-10773-2018>, 2018.

Wang, J., Chagnon, F. J. F., Williams, E. R., Betts, A. K., Renno, N. O., Machado, L. A. T., Bisht, G., Knox, R., and Bras, R. L.: Impact of deforestation in the Amazon basin on cloud climatology, *Proceedings of the National Academy of Sciences*, 106, 3670–3674, <https://doi.org/10.1073/pnas.0810156106>, 2009.

Zha, Q., Aliaga, D., Krejci, R., Sinclair, V. A., Wu, C., Ciarelli, G., Scholz, W., Heikkinen, L., Partoll, E., Gramlich, Y., Huang, W., Leiminger, M., Enroth, J., Peräkylä, O., Cai, R., Chen, X., Koenig, A. M., Velarde, F., Moreno, I., Petäjä, T., Artaxo, P., Laj, P., Hansel, A., Carbone, S., Kulmala, M., Andrade, M., Worsnop, D., Mohr, C., and Bianchi, F.: Oxidized organic molecules in the tropical free troposphere over Amazonia, *National Science Review*, 11, nwad138, <https://doi.org/10.1093/nsr/nwad138>, 2024.

Version 1:

Reviewer comments:

Reviewer #1

(Remarks to the Author)

I would like to express my appreciation to the authors for addressing all my suggestions, recommendations and concerns so thoroughly.

Thus, I recommend the publication of this manuscript.

Reviewer #2

(Remarks to the Author)

I thank the authors for their clear responses to all the comments raised by the reviewers. I am now satisfied.

Reviewer #3

(Remarks to the Author)

I think the manuscript has improved, but I am still not a 100% convinced that the modeling part of this study adds much new to the existing knowledge on short-lived climate forcers. Since the transport of BVOCs to the UT is the main point of the work, this kind angle should come across more in the modeling part (I know this is discussed, but it could be highlighted/investigated more). Still, the flight data on BVOCs is really impressive and nicely shown/discussed.

Minor comments:

-L158 says "Joe et al. (2021)", should be Jo et al. (2021)

-L193–196, something seems off with the sentence

-L199–200 I'd suppose these are most relevant for climate only if the particles are transported back down with convective downdrafts/subsidence and one must assume that the newly formed particles survive this transport to a warmer atmosphere

-L219: Just a note: MT (as in middle troposphere) can be confused with monoterpenes, often also abbreviated with MT (I got confused for a second)

-Figures 2&3, 5 lack a–c labels

-Figure 4: perhaps note different scales of vapor mixing ratios shown with color bars. You could also choose a different color for altitude to differentiate the totally different parameter displayed.

-L366: Where can this ozone effect be seen? Is there a plot somewhere you should refer the reader to?

-L375: consider rephrasing "radiative energy budget"

-L380-L381: consider rephrasing "radiative warming forcing"

-L378–379: The sentences "Given that reducing BVOC emissions..." and "It may be tempting ..." do not link well here

Response to referees

General comments.

We thank the reviewers and Editor for their helpful feedback. All points raised have been addressed, and the responses documented in detail below. There were three points that were brought up by more than one reviewer. Since these common themes can be considered particularly important, we comment on them at the beginning before proceeding to the point-by-point replies.

- 1) Two reviewers are familiar with previous airborne measurements done in the vicinity of the city of Manaus in which a NO_x rich city plume was tracked across a short section of rainforest and the resultant ozone photochemistry monitored. While certainly interesting, we would like to point out that these conditions are confined to a relatively small area of rainforest downwind of Manaus. They are not typical of the pristine rainforest, which characteristically has very low NO mixing ratios. The aim of our study was to investigate the unperturbed rainforest chemistry, as the long range of our aircraft allowed us to cover vast areas of pristine rainforest throughout the Amazon basin. We now make this clearer both in the text and with a figure of the respective areas of operation. We also include the references suggested by the reviewers to note that indeed small regions of high NO_x and vigorous photochemistry do exist near to cities. However, our aim is to characterize the more common and widespread chemical conditions of the rainforest region.
- 2) Two reviewers requested supporting evidence that deep convection was occurring and causing the vertical concentration profiles presented. In the field, it was clear from the daily meteorological reports, satellite information and flight planning that the entire Amazon basin was experiencing large-scale vigorous convection both day and night. Moreover, the presence of a short-lived chemical compound such as isoprene (lifetime 1-2 hours) at 10-12 km is only possible through deep convection. We agree that this information was missing in the paper. In order to completely substantiate this interpretation, we now provide both small-scale (individual cell) and large-scale basin-wide imagery showing widespread convection and enhanced isoprene in outflow regions.
- 3) Two reviewers questioned whether the modelling work authentically simulated deforestation. Indeed, we agree entirely with the reviewers that it does not, as many more parameters would have to be changed. However, this was not the aim of our simulations, and we apologize if we implied otherwise. Instead, we wanted to highlight the atmospheric role of the biogenic VOCs by varying them substantially in the model. To make it clearer that this is a sensitivity study and not an attempt to simulate the multiple effects of deforestation, we now decrease and increase the BVOC emission, noting the impact on OH, O₃ and radiative forcing. We make the BVOC decrease in the model match the decrease we actually measured over deforested areas, and the increase matching a CO₂-doubling scenario to maintain the connectivity and realistic conditions in the paper. Thus, we now perform three sensitivity studies with adjusted BVOC emissions from the Amazon rainforest: (a) 50% decrease, (b) 75% decrease and additionally now in the revised version (c) 50% increase. We now make it very clear in the text we are not simulating deforestation. We present the updated Figure 5 of the manuscript in Figure R1.

Figure R1. Updated version of Figure 5 from the manuscript. We replaced the sensitivity simulation of the 50% globally decreased BVOC emissions with a simulation locally increasing BVOC emissions by 50%. The figure provides annual averaged changes in ozone and OH mixing ratios over the Amazon basin, as well as the global aerosol direct radiative effect from changes in BVOC emissions. More details on the Figure will be provided in the revised manuscript.

Response to the Reviewer #1

General comment:

The manuscript is mainly based on discussions of BVOC and ozone measurements from the CAFE-Brazil experiment during December 2022–January 2023. Measurements in the Amazon rainforest have become even more important considering the urgent scenario we find ourselves in regarding reducing and predicting climate change and the extreme events resulting from that. The paper has a very good description of the measurements and a valid scientific question to be further investigated. My main concern is related ensuring the scientific contribution of the paper is well supported. The link-approach and logic sequence in developing and discussing the papers conclusion are not clear and well aligned. There are some technical and scientific issues that need to be addressed for the authors before I can give my full recommendation for publishing the manuscript.

The authors sincerely thank the reviewer for the valuable comments and constructive suggestions. In response to the feedback, we have carefully revised the manuscript to enhance the clarity of the scientific contribution and the logical flow of the study's objectives and conclusions. We have specifically worked to clarify the link between our measurements, the model calculations and the scientific questions addressed, as well as to ensure a clear and cohesive development of the conclusions. We believe the revisions have strengthened the manuscript, addressing both the technical and scientific concerns raised. A detailed point-by-point response to each of the reviewer's comments is provided below.

Specific comments/questions:

- I recommend the authors work to define what it the main goals/contributions they want from this paper:

(i) deep convection events transporting BVOCs to the troposphere and how they stay there due to residual boundary layer and low oxidant variabilities, so in the early morning those contributions trigger NPF formation. The authors can show this using the CAFE-Brazil data and using the model to investigate how much this “base scenario” would change if the BVOCs have reduced emissions. To do that the authors need to first show the model is minimally capable of reproducing background conditions for gases and particles. Once the model can reproduce BVOCs concentrations (evaluate process with observational data), then the sensitive modeling experiment can be performed.

Reply. Thank you for your valuable comments. The main objective is to understand the transport and chemistry of BVOCs to the upper troposphere during deep convection under background conditions. As we note in the manuscript already, during the daytime, most of the highly reactive BVOCs, such as isoprene, are quickly oxidized. However, during the nighttime, these compounds can be transported from the lower troposphere and accumulate in the upper troposphere due to the absence of photochemical reactions. After sunrise, in the presence of lightning-generated NO_x and OH radicals, these BVOCs can contribute to NPF formation. Subsequently, we performed the sensitivity study using EMAC model to investigate the sensitivity of Amazonian atmospheric chemistry and radiative effects on changes in BVOC emissions.

The motivation of sensitivity study stems from the observation of elevated levels of BVOCs in the UT during convection, raising the question how changes in the BVOC emission at the surface will impact the atmospheric chemistry from the boundary layer to the UT (9-14 km). Additionally, a low-altitude flight (900 m) contrasting pristine forested and deforested areas provided us an additional motivation to perform a BVOC sensitivity study using the global atmospheric chemistry model EMAC. Overall, this study aims to investigate the atmospheric impacts of the Amazon rainforest's current BVOC emissions and explores impacts of potential changes of these emissions. We have revised the manuscript to clarify our objectives.

Following the advice of the reviewer, we have provided more evidence that the model is capable of reproducing background concentrations for gases and particles. We perform an evaluation of simulated ozone, OH, and organic aerosol using observations from the ATom missions, taking place in four different seasons from 2016 to 2018, covering background conditions in the Atlantic, Pacific, Southern Ocean, continental US, and Canada (see Figure R2-R4). Additionally, we show a comparison of simulated monoterpene and isoprene mixing ratios to the measurements at the ATTO tower above the canopy (320 m) during the CAFE-Brazil campaign in December 2022 and January 2023 (Figure R5). Moreover, we point out to additional references, where the EMAC model with similar setups was evaluated, with regard to aerosols¹⁻⁴, ozone and OH⁵. We also now provide these references in the revised manuscript. As the base model has good representation of background gases and aerosols, the sensitivity studies when we raise and lower the BVOC emissions, noting the impact on OH, O₃, and the radiative effect are then justified.

The changes in the revised manuscript are- (line no. 102-107)

“A low-altitude flight (900 m) contrasting pristine forested and deforested areas provided us an additional motivation to perform a BVOC sensitivity study using the global atmospheric chemistry model EMAC. This study aims to investigate the atmospheric impacts of the Amazon rainforest's current BVOC emissions and explores impacts of potential changes of these emissions. The model was then used to assess the atmospheric chemistry of BVOCs over the Amazon rainforest and the impact of changes associated with their modulation.”

The changes in the revised manuscript are- (line no. 263-273)

The EMAC model simulations were performed to investigate the sensitivity of atmospheric chemistry above the Amazonia basin and the radiative effects to changes in BVOC emissions. The motivation of the sensitivity study is the elevated levels of BVOCs measured in the UT associated with convection and addressing how changes in the BVOC emission at the surface will impact the atmospheric chemistry from the boundary layer to the UT (9-14 km). Flight 06 provides notable motivation for this sensitivity study. In this flight, in fact, the measurements documented BVOCs over deforested areas converted to pasture, and contrasted to nearby pristine forests sampled at relatively low altitudes (~900 m) in the boundary layer over Rondônia, Brazil. Figures 4a-d depict the variations in the altitude and mixing ratios of isoprene, isoprene-OP and monoterpenes along flight 6 over forested and deforested regions, respectively.

The changes in the revised manuscript are- (line no. 302-306)

“However, surface monoterpene and isoprene mixing ratios were evaluated with measurements from the ATTO tower, and simulated ozone, OH and organic aerosol were additionally

evaluated with background aircraft observations, showing good agreement (supplementary Figures 9-12). Moreover, previous studies have evaluated the EMAC model with similar setups for aerosol⁵⁷⁻⁶⁰, ozone and OH⁵⁵.”

Figure R2. Comparison of vertical profiles of observed and simulated O₃ mixing ratios during the NASA Atmospheric Tomography (ATom) mission in the four sub-campaigns at different seasons: a) ATom-1 (July/August 2016), b) ATom-2 (January/February 2017), c) ATom-3 (September/October 2017), and d) ATom-4 (April/May 2018). The box-whisker plot represents the mean (circle), median, lower quartile, upper quartile and the 5th and 95th percentiles for 1km bins. The observation and model data are represented by blue and red color respectively. Observations are taken from Elkins et al. (2019)⁶.

Figure R3. Comparison of vertical profiles of observed and simulated OH mixing ratios during the NASA Atmospheric Tomography (ATom) mission in the four sub-campaigns at different seasons: a) ATom-1 (July/August 2016), b) ATom-2 (January/February 2017), c) ATom-3 (September/October 2017), and d) ATom-4 (April/May 2018). The box-whisker plot represents

the mean (circle), median, lower quartile, upper quartile and the 5th and 95th percentiles for 1km bins. The observation and model data are represented by blue and red color respectively. Observations are taken from Brune et al. (2021)⁷.

Figure R4. Comparison of vertical profiles of observed and simulated organic aerosol mass concentrations during the NASA Atmospheric Tomography (ATom) mission in the four sub-campaigns at different seasons: a) ATom-1 (July/August 2016), b) ATom-2 (January/February 2017), c) ATom-3 (September/October 2017), and d) ATom-4 (April/May 2018). The box-whisker plot represents the mean (circle), median, lower quartile, upper quartile and the 5th and 95th percentiles for 1km bins. The observation and model data are represented by blue and red color respectively. Observations are taken from Jimenez et al. (2019)⁸.

Figure R5. Comparison of the time series of modeled and observed (a) isoprene and (b) monoterpene mixing ratios at the Amazon Tall Tower Observatory (ATTO, 320 m), and model simulated values. The blues line shows observations, while the red line shows the model output extracted from the grid cell covering 240 to 580 m.

(ii) Only focusing on the observational side for BVOCs, O₃ and NO_x. In this case, I strongly recommend the authors add to their analysis the set of important data from the GoAmazon 2014/2015 experiment and the data from ATTO tower since 2013 (ozone, NO_x and VOCs).

This would give more robust statistics to evaluate future consequences for a scenario with BVOC reduction, it's important to analyze how those compounds change seasonally as well. It's not clear that the conclusions reached in this paper are true generally, or just a result of some seasonal or annual variation in the two months used by the authors. Additional observations would alleviate these concerns.

Reply. We take the reviewer's point.

However, we note that the GoAmazon 2014/15 campaign covered only a small NO_x-rich region downwind of Manaus. This region is not representative of the wider pristine Amazon rainforest, which is characterized by low NO_x. The objective of our campaign was to study the impact of convection over the pristine Amazon rainforest. Nevertheless, in order to address the concerns of seasonal and annual variation, we now extend the study using the model to explore these changes. We also note the previously reported seasonal changes in BVOC emissions (Yáñez-Serrano et al. 2015)⁹ and that “the transition season from dry to wet and the beginning of the wet season is generally the period of the strongest intensity of convection” (Machado et al. 2004)¹⁰. We also refer to the Go Amazon literature to make the distinction between chemical regimes.

Specifically, the vertical profiles of relative changes in O₃ and OH, as shown in Figure 5, are now simulated for the entire year, rather than being limited to the CAFE-Brazil campaign. Additionally, we provide seasonal assessments in the supplementary data presented in Figure R6 here, which highlights the seasonal impacts of changes in BVOC emissions. Indeed, it suggests that the time frame of the CAFE-Brazil campaign is not fully representative of the entire year. In particular, the impact on OH and ozone exhibits strong seasonal variability at the surface. In the dry season, higher BVOC emissions lead to an increased impact on the boundary layer. However, the reduced convective activity during the dry season leads to a diminished impact in the UT. Despite this, the CAFE-Brazil campaign remains representative of the influence of BVOCs in the UT due to the strong convective transport during the wet season and the transition from dry to wet periods. This underscores the importance of seasonal and annual assessments in understanding the broader implications of BVOC emissions on atmospheric chemistry.

Figure R6. Seasonal analysis of the relative changes in ozone and OH, along with the global direct aerosol radiative effect for the three sensitivity studies, analogously to Figure 5. The panel labels represent: (a1–a3) wet season, (b1–b3) June/July, (c1–c2) dry season, and (d1–d3) December/January. The wet season is defined from February to May and the dry season from August to November, according to The seasons are classified according to Andreae et al. (2015)¹¹ and Pöhlker et al. (2016)¹².

The changes in the revised manuscript are given below– (Line no. 77-80)

“Although a previous study reported the strong seasonal variation in the BVOC emissions at the surface over the Amazon rainforest²⁸, Machado et al. (2004)²⁹ reported the strong intensity of convection during the dry-to-wet transition periods and the onset of the wet season.”

(Line no. 311-326)

“The annual and seasonal changes in the Amazonian region over 0-20 km altitude, relative to the base case, for the oxidants OH radicals, O₃, and the global radiative effect due to aerosol, are illustrated in Figure 5 and Supplementary Figure 13, respectively. Each panel in Figure 5 indicates the relative change across the atmospheric column. The results show that BVOCs strongly influence the oxidation capacity of the atmosphere, even up into the UT. Annually, the strongest effects on OH (changes of 300% relative to the base case, when BVOC emissions were lowered to 25%), occurred in the boundary layer at 0-3 km altitude. The highest relative change in OH (~400 %) in the LT (0-3 km) during the dry season suggests a higher loss rate of OH, driven by elevated BVOC emissions over the Amazon basin during this period (Supplementary Figure 13). In the mid-troposphere (5-10 km), a 75% decrease in BVOC emissions leads to a 100-150% relative increase in OH mixing ratios, On the other hand, OH only decreases by 15-40% after increasing BVOC emissions by 50% at these altitudes. This relative change in OH after reducing and increasing BVOC emission shows a non-linear dependency (Figure 5b). An increase in BVOC emissions can lead to a reduction in OH levels; however, the sensitivity of OH to increases in BVOC is buffered, as other sinks and competing processes also play a significant role⁴⁷.”

- How did the authors define the convection events (Convective transport of air from low altitudes)?

Reply. Convection is the vertical movement of air, driven by temperature differences, a key process for transferring heat, moisture and trace compounds throughout the depth of the troposphere. In this study, the deep convection events were identified based on the observed elevated levels of short-lived primary BVOCs, such as isoprene and monoterpenes, in the upper troposphere (see Figure R7). These compounds, emitted from terrestrial sources, typically have atmospheric lifetimes of 1-2 hours, making their transport to such high altitudes unlikely without deep convective activity. Meteorological data shows clear evidence of large-scale convection over the entire Amazon basin, for example, the cloud top brightness temperature variation (see Figure R8). A very low cloud-top brightness temperature (often below -50°C to -70°C) indicates deep convection.

To address this, we have revised the manuscript to include a figure illustrating a specific flight where high isoprene mixing ratios were measured in the upper troposphere associated with a convection event, providing direct evidence of convective transport events (See Figure R7). We have also included the cloud top brightness temperature to identify the convective system and HALO aircraft location with high isoprene values (Figure R8). The revised sentence in the manuscript highlights this finding to strengthen the discussion on convective events.

The revised sentence is (Line no. 144-147)

“The elevated mixing ratios of isoprene, isoprene-OP and monoterpenes in the UT provide clear evidence of extensive rapid transport to this altitude associated with deep convection. Supplementary Figure 3 depicts an example of the deep convection of isoprene in the UT for Flight 10.”

Figure R7. The time series of isoprene observed during flight 10, with the transparent strip highlighting periods of elevated isoprene levels associated with deep convection.

Figure R8. The cloud-top brightness temperature is representing the deep convective system during flight 10. Infrared satellite image (GOES-16, band 13: 10.3 μm ; <https://ftp.cptec.inpe.br/goes/goes16/retangular/ch13/2022/12/>) indicating the approximate cloud-top brightness temperatures at 19:40 UTC. Temperatures below -40°C are colored. The yellow circle indicates the HALO aircraft location at 19:44 UTC (15:44 Local time), and the colors correspond to the temperatures measured during the flight. The map in this figure was made with Natural Earth. Free vector and raster map data @ naturalearthdata.com

- The authors should discriminate better between urban and forest areas during the flight pathway and explore more the differences in the measurements in these areas. The area where the CAFE-Brazil experiment was developed has a city (almost 12km^2) with more than 2 million people and it has a huge impact on the photochemistry and oxidation capacity around the forest. The 2 following papers about the dynamics of O_3 production in the Amazon rain forest:

Reply. The authors agree with the reviewer that urban plumes can significantly influence photochemistry downwind of the city of Manaus, with a 2 million population. Shilling et al. (2018)¹³ showed the impact of such urban plumes on nearby forested areas at altitudes of 500 to 1000 m. In this study, they mainly covered the regions immediately surrounding the city of Manaus, an area of circa 15,000 km², which accounts for about 0.6 % of the area covered during the CAFE-Brazil campaign (approximately 2.7 million km²).

However, the area covered during the CAFE-Brazil campaign was predominantly under pristine condition, by far exceeding the polluted/urban conditions. In fact, along the flight paths, we mostly observed background levels of anthropogenic benzene, xylene, and acetonitrile, with very few spikes of benzene recorded in the lower troposphere throughout the campaign. To address this point, the spatial distribution of the anthropogenic compound benzene along the flight track is now shown in Figure R11, to confirm the limited influence of anthropogenic VOCs on our measurements during the CAFE-Brazil campaign. Additionally, to minimize the influence of the urban plume, the ambient air measurements were initiated only after reaching a certain altitude (3 to 5 km) after takeoff and stopped ambient air measurements around 1 to 3 km prior to the landing.

For reference, Figures R9 and R10 illustrate the area covered during the GoAmazon2014/5 and CAFE-Brazil campaigns, along with the vertical profile of aromatics measured over the yellow-shaded area (pink) and the rest of the regions (green) during the CAFE-Brazil campaign. The median value of benzene (24 ppt) in the boundary layer was measured near the LOD (12 to 22 ppt for different flights).

Figure R9. Flight tracks of the HALO aircraft during CAFE-Brazil campaign (red lines) and the yellow polygon shows the approximate area covered during GoAmazon2014/5 campaign. The underlying image is from Google Earth (Map Data: Google, @ 2021, SIO, NOAA, U.S. Navy, NGA, GEBCO Landsat / Copernicus INEGI IBCAO).

In the updated manuscript, we have included the statement given below- (Line no. 80-86)

“This campaign primarily focused on BVOC measurements over pristine areas to minimize the influence of urban plumes and biomass burning (which was less prevalent during this period)³⁰⁻³². Previous studies have shown that urban plumes can significantly influence photochemistry and the oxidation capacity in the downwind forest regions^{33,34}. However, the low median value of benzene (24 ppt) in the boundary layer measured near the limit of detection (LOD, 12 to 22 ppt for different flights) further indicates that urban emissions were not a significant source in the area covered during this study (Supplementary Figure 1).”

Figure R10. The comparison of the vertical profile of benzene was measured near the downwind region of Manaus (yellow polygon in Figure R9) and the rest of the locations covered during CAFE-Brazil. Points over pristine rainforest $n \sim 6000$, points in the Manaus downwind region $n = 241$.

Figure R11. The flight tracks during the CAFE-Brazil campaign over the Amazon rainforest color-coded by benzene mixing ratios. Satellite picture data in this figure obtained from <https://wvs.earthdata.nasa.gov>, NASA Worldview Snapshots.

Shilling, J. E.; Pekour, M. S.; Fortner, E. C.; Artaxo, P.; de Sá, S.; Hubbe, J. M.; Longo, K. M.; Machado, L. A. T.; Martin, S. T.; Springston, S. R.; Tomlinson, J.; Wang, J. Aircraft observations of the chemical composition and aging of aerosol in the Manaus urban plume during GoAmazon 2014/5. *Atmos. Chem. Phys.* 2018, 18,10773–10797.

Nascimento, J. P.; Barbosa, H. M.; Banducci, A. L.; Rizzo, L. V.; Vara-Vela, A. L.; Meller, B. B.; Gomes, H.; Cezar, A.; Franco, M. A.; Ponczek, M.; Wolff, S.; Bela, M.; Artaxo, P. Major Regional-Scale Production of O₃ and Secondary Organic Aerosol in Remote Amazon Regions from the Dynamics and Photochemistry of Urban and Forest Emissions. *Environ. Sci. Technol.* 2022, 56, 9924– 9935, DOI: 10.1021/acs.est.2c01358

- The authors should show evidence of the claimed deep outflow events and associate them with the BVOCs (before, during and after the event).

Reply. We have included isoprene time series (for flight 10) covering a convective event that represent the BVOC levels before, during and after the convective events (see the above Figure R7-R8).

The sentences included in the updated manuscript- (Line no. 144-147)

“The elevated mixing ratios of isoprene, isoprene-OP and monoterpenes in the UT provide clear evidence of extensive rapid transport to this altitude associated with deep convection. Supplementary Figure 3 depicts an example of the deep convection of isoprene in the UT for Flight 10.”

- The follow paragraph has a reference that seems unrelated to its content: “To assess the full atmospheric effects of deforestation, the chemical and physical impacts of BVOC must be characterized and understood.”

Reply. We have revised the references. We have added new references, “Ganzeveld and Lelieveld, 2004¹⁴ and Weber et al., 2022¹⁵.”

(Line no. 71-73)

“To assess the full atmospheric effects of deforestation and climate change, the chemical and physical impacts of BVOC must be characterized and understood^{26,27}.”

- Line 98, the authors should define what “MR” is.

Reply. We mentioned it in Line 77 when we used first time. To avoid confusion, we have now replaced “MR” with “mixing ratios” throughout the manuscript.

- Figure 02: The image needs to be improved. The resolution is very poor making it hard to read the inset plots. If possible, vector images would be much better. The inset plots right plot with the day (red)/ night (blue) are confusing to me because the y axis doesn't go to lower than 9km and BVOC values at the PBL high are bigger in the morning due to the emissions being correlated with solar radiation. The authors should add metadata that describes how the plots from the flight were made: eg include range, hours, location, days, etc.

Reply. Thank you for your valuable suggestions. We have revised all plots in the manuscript to vector images for improved resolution.

Regarding the inset plots in Figure 2, we used data for higher altitudes (9–13 km) to highlight the differences in the mean values of BVOC in the upper troposphere during daytime and nighttime. Including the complete vertical profile would obscure these differences due to the larger mixing ratios in the lower troposphere. Additionally, as mentioned in the manuscript, nighttime flights were not conducted within the boundary layer (not possible under VFR flight rules in the dark), making it statistically challenging to compare daytime and nighttime averages at lower altitudes (0–1 km).

However, to address the concerns, we have now added the complete vertical profiles of BVOCs for daytime and nighttime in the supplementary information. It is important to note that the nighttime dataset at the lowest altitude contains only 51 data points, with ~40 measured between 18:00 and 19:00. The revised figure caption now includes metadata describing the range, hours, location, and days for the flight measurements (see Figure R13).

Figure R12. Vertical profiles of isoprene, its oxidation products (OP) and monoterpenes during the CAFE Brazil campaign. Gray dots are 1 min-averaged data, while mean values are represented by a solid green line (1 km bin); the shaded light-green area shows the standard deviation. Note the mean value is calculated only for values above the LOD. The black square with error bars represents the mean value of ATTO data measured during the flight days. The inset plots show the mean vertical profiles of BVOCs at higher altitudes (9-14 km) during day- and nighttime in red and blue, respectively.

Figure R13. Vertical profiles of isoprene, its oxidation products, and monoterpenes during the CAFE-Brazil campaign over the Amazon rainforest during the day (upper panel) and night (lower panel). The profiles are binned to a 1 km vertical resolution grid, with the number of data points used for each box displayed above it. The boxes represent the 25th–75th percentiles, with red (daytime)/blue (nighttime) lines indicating the median while red (daytime)/blue (nighttime) dots indicate the mean values. Whiskers extend to the lowest and highest data points within $1.5 \times$ Interquartile Range (IQR), the data point beyond this range is considered outlier. Black dots represent outliers. A total of 15 days of flight (~ 125 hr) data was used, with daytime data spanning 06:00 to 18:00 local time (~ 100 hr) and nighttime data (~ 25 hr) covering 18:00 to 06:00 local time. Only data above the LOD is used for this box-whisker plot. The number of data points for these BVOCs differs due to their respective LOD values.

The sentences included in the updated manuscript- (Line no. 116-117)

“The vertical profiles of isoprene, isoprene OP, and monoterpenes for all flights are shown in Figure 2 and Supplementary Figure 2.”

(Line no. 147-150)

“The aircraft sought and captured multiple deep convection outflow events in the UT during day and night, although mixing ratios for all three species at ~ 12.5 km were consistently higher at night (see inset Figure 2 and Supplementary Figure 4).”

- Figure 03: Replacing this with a vector image would be very helpful in order to easily compare the difference in the measurements between the heights. Additionally, the region the measurements are taken needs to be specified. Is it urban or forest? If the airplane is moving how can data be compared with a stationary source?

Reply. Thank you for your suggestions. We have replaced the figure with the vector image. As stated above, we flew over the pristine forested regions, as shown in Figure 1. The mixing ratios of BVOCs measured in the boundary layer during daytime over the pristine forested region are comparable to the values measured over the ATTO tower. In the diel cycle plot, we implicitly assume that these BVOCs are at similar levels throughout the forested regions of Amazonia. In this section, our aim is to explain the diel variation of BVOCs at different altitude

regions. Remarkably, the diurnal profiles of BVOCs show completely opposite patterns at the surface and in the upper troposphere of the Amazon rainforest. Their mixing ratios are higher near the sources during the daytime. In contrast, in the upper troposphere, their levels mainly depend on convective transport and photochemistry. The mixing ratios of these BVOCs were higher during nighttime in the absence of photochemistry.

Figure R14. Diel cycles of isoprene, its oxidation products (OP) and monoterpenes for three different altitudes regimes of 320 m (left side y-axis) at ATTO, 3-9 km, and 9-14 km (right side y-axis) during the CAFE-Brazil campaign.

- Lines 206-207 “It is important to note that deep convection events were widespread and frequent throughout the Amazon basin both by day and night.”. Can the authors show data evidencing this premise? It would enrich the paper discussion and scientific contribution for the community, especially due the fact all these events are yet not well defined and understood. A field campaign such as CAFE-Brazil is a great opportunity to help on better understanding those processes.

Reply. Thank you for the suggestion. We have now included satellite data plots of cloud top brightness temperature in the revised manuscript to demonstrate convective events. These include a small-scale example of the aircraft crossing the convective outflow and encountering high isoprene and also a larger-scale image exemplifying the regional widespread convection (Figure R8). Additionally, we have revised the manuscript to include sentences emphasizing the role of convection in vertical transport during daytime and nighttime that can transport BVOCs and influence chemistry in the UT (Figure R15). We do have a separate manuscript in preparation which examines individual convection events in great detail; however, in this paper, we provide an overview of the entire CAFE-Brazil campaign.

Figure R15. The cloud-top brightness temperature represents deep convective systems during the day (upper panel) and night (lower panel) for flight 10. Infrared satellite image (GOES-16, band 13: 10.3 μm ; <https://ftp.cptec.inpe.br/goes/goes16/retangular/ch13/2022/12/>) indicating the approximate cloud-top brightness temperatures at 19:40 and 00:10 UTC. Temperatures below $-40\text{ }^\circ\text{C}$ are colored. The map in the figures were made with Natural Earth. Free vector and raster map data @ naturalearthdata.com.

The revised sentence in the updated manuscript-(Line no. 233-235)-

“It is important to note that deep convection events were widespread and frequent throughout the Amazon basin both by day and night (Supplementary Figure 7).”

- Figure 04: This figure needs to be rethought because is very confusing, the color scale of the 2 different flights is not the same. The brown-green for deforested and forest areas are not very obvious (maybe call it green/tan?). The figures should show a bigger picture of the area where the flights took place the urban/forest/deforested area could be more highlighted. It might be easier to understand this data and support the authors point if it was a scatter plot showing

altitude vs atmospheric constituent (isoprene, monoterpene, etc) and the points were colored green or brown for forested/deforested.

Reply. We agree with the reviewer that the color scales differ between flight 6 and flight 17. Moreover, Flight 17 did not cover the lower altitudes (<1000 m) over the forested regions and was thus not used to define the lower range for the BVOC sensitivity study. Therefore, to simplify and improve the graphic, we removed the Flight 17 plots from Figure 4. In doing so, we have increased the area for flight 6 as per the reviewer’s suggestion (Figure R16). Since we did not cover the deforested region at high altitudes, it would be challenging to depict the vertical profile of BVOCs using different colors (e.g., green for forested and tan for deforested regions). We have also updated the figure caption to clarify that green represents forested regions, while tan represents deforested regions, as suggested by the reviewer.

The modified caption is (Line No. 308-310)-

“Figure 4. Flight tracks of flight 06 with color bar a) altitude in meters and mixing ratios of b) isoprene c) isoprene OP and d) monoterpenes over the Amazon rainforest. Green and tan colors on the map show the forested and deforested regions, respectively.”

Figure R16. Flight tracks of flight 06 with color bar a) altitude in meters and mixing ratios of b) isoprene c) isoprene OP and d) monoterpenes over the Amazon rainforest. Green and tan colors on the map show the forested and deforested regions, respectively. Satellite picture data in this figure obtained from <https://wvs.earthdata.nasa.gov>, NASA Worldview Snapshots.

- Line 295-297 “Interestingly, the net changes in O₃ vary strongly with altitude. When Amazonian BVOC emissions are reduced by 75% in the model, O₃ increases by circa 10% in the LT (0-4 km) but decreases by about the same amount in the UT (between 13-16 km).”

The authors should clarify in which location the O₃ shows the increase (10%) due to the BVOCs reduction. In forest regions such as ATTO tower region, the O₃ production is limited by NO_x emissions, so a reduction on BVOCs is unlikely to be the cause of the O₃ increase.

Variations in NO_x are more aligned to changes in O₃ concentration on NO_x-limited regions. I recommend the authors:

- (i) Distinguish the regions when the O₃ was increased
- (ii) Be more specific with the time range related to the increase.
- (iii) Compare ozone observational data with simulations (without the BVOCs reduction), so the analyses would be more realistic when comparing ozone concentrations.

Reply. We agree with the reviewer that ozone production in forested regions, such as around the ATTO tower, is NO_x-limited. Thus, ozone production will not decrease significantly with reduced BVOC emissions. However, ozone loss in the boundary layer is strongly impacted by the direct reaction of ozone with reactive BVOC. In Figure R1, elevated ozone levels in the lower troposphere do not indicate photochemical NO_x-catalyzed ozone production; rather, it shows that surface ozone is suppressed by direct reaction with BVOC, which, when lowered in our sensitivity tests, causes the ozone to increase. Multiple studies have reported that BVOCs, particularly monoterpenes, are highly reactive to ozone^{16–18}. Therefore, if these BVOCs are reduced by 50–75%, the ozone sink would diminish, leading to higher ozone levels.

- (i) Distinguish the regions when the O₃ was increased

Ozone increases over the whole Amazonian region by reducing 50 and 75 % BVOCs. To make this point clearer, additionally, we have now added the reduction in ozone due to an increase in BVOCs by 50 % over the Amazon rainforest. We have included a map of changes in ozone mixing ratios in the boundary layer due to changes in BVOC emissions over the Amazon rainforest in the supplementary material (see below Figure R17).

The modified paragraph in the updated manuscript is (Line no. 332-348)-

“Interestingly, the net annual changes in O₃ vary strongly with altitude. When Amazonian BVOC emissions are reduced by 75% in the model, O₃ increases by circa 22 % in the LT (0-4 km) averaged over the Amazon basin (Figure 5a), with local increases exceeding 30% (Supplementary Figure 14), but decreases by 10% in the UT (between 13-16 km). Seasonally, the strongest influence on O₃ at the surface is observed during the dry season (>25% relative change), while this impact is more pronounced (~ -10%) in the UT during the wet season and the transition from dry to wet periods (Dec–Jan, Supplementary Figure 13). This indicates that the significant convective transport of BVOCs during the wet seasons plays a pivotal role in the production of O₃ in the UT. The O₃ increase in the lower troposphere can be attributed to lower rates of direct BVOC-O₃ reactions, particularly for monoterpenes that react faster with O₃ than isoprene⁶². Therefore, we infer that BVOCs act mainly as the sink for O₃ (Supplementary Figure 14) because the rainforest boundary layer is a NO_x poor environment, where photochemical O₃ production is limited by extremely low NO^{50,51}. In contrast, the high-altitude regions are enriched in NO_x by lightning, and the reduction in O₃ in the UT due to the reduction in BVOC levels reveals that O₃ production is limited by BVOCs, and that lower surface emissions mean less BVOC is available aloft to drive photochemical O₃ production.”

Figure R17. The change in ozone after reducing a) 50 and b) 75 % and c) increasing 50% of BVOCs emission in the boundary layer of Amazon rainforests. The square shows the area referred to as the Amazon basin in simulations. The map in this figure was made with Natural Earth. Free vector and raster map data @ naturalearthdata.com.

(ii) Be more specific with the time range related to the increase.

The vertical profile presented in Figure R1 represents the annual change, while the seasonal variation is also included (see Figure R6). Additionally, the above maps are annually averaged.

We also provided this information in the updated manuscript (Line no. 311-314).

“The annual and seasonal changes in the Amazonian region over 0-20 km altitude, relative to the base case, for the oxidants OH radicals, O₃, and the global radiative effect due to aerosol, are illustrated in Figure 5 and Supplementary Figure 13, respectively. Each panel in Figure 5 indicates the relative change across the atmospheric column.”

(iii) Compare ozone observational data with simulations (without the BVOCs reduction), so the analyses would be more realistic when comparing ozone concentrations.

Thank you for your suggestion. In order to substantiate the model’s ability to simulate ozone in this region, we now provide a comparison using data from the Amazon-adjacent ATom campaign, where the simulations showed good agreement with the observational data (see Figure R2-R4). In addition, we refer to Jöckel et al. (2016)⁵, where EMAC-simulated ozone and OH were evaluated on a global scale.

The included sentences in the updated manuscript are (Line no. 302-306)-

“However, surface monoterpene and isoprene mixing ratios were evaluated with measurements from the ATTO tower, and simulated ozone, OH and organic aerosol were additionally evaluated with background aircraft observations, showing good agreement (supplementary Figures 9-12). Moreover, previous studies have evaluated the EMAC model with similar setups for aerosol⁵⁷⁻⁶⁰, ozone and OH⁵⁵.”

- The authors should consider the finds in the 2 following papers about the dynamics of O₃ production in the Amazon rain forest:

Shilling, J. E.; Pekour, M. S.; Fortner, E. C.; Artaxo, P.; de Sá, S.; Hubbe, J. M.; Longo, K. M.; Machado, L. A. T.; Martin, S. T.; Springston, S. R.; Tomlinson, J.; Wang, J. Aircraft observations of the chemical composition and aging of aerosol in the Manaus urban plume during GoAmazon 2014/5. *Atmos. Chem. Phys.* 2018, 18,10773–10797.

Nascimento, J. P.; Barbosa, H. M.; Banducci, A. L.; Rizzo, L. V.; Vara-Vela, A. L.; Meller, B. B.; Gomes, H.; Cezar, A.; Franco, M. A.; Ponczek, M. Major Regional-Scale Production of O₃ and Secondary Organic Aerosol in Remote Amazon Regions from the Dynamics and Photochemistry of Urban and Forest Emissions. *Environ. Sci. Technol.* 2022, 56, 9924– 9935, DOI: 10.1021/acs.est.2c01358

Reply. We thank the reviewer for these suggestions. Both references are now included in the updated manuscript (**Line no. 82-86**).

“Previous studies have shown that urban plumes can significantly influence photochemistry and the oxidation capacity in the downwind forest regions^{33,34}. However, the low median value of benzene (24 ppt) in the boundary layer measured near the limit of detection (LOD, 12 to 22 ppt for different flights) further indicates that urban emissions were not a significant source in the area covered during this study (Supplementary Figure 1).”

- Line 297-300 “The O₃ increase close to the surface can be attributed to lower rates of direct BVOC- O₃ reactions, particularly for monoterpenes that react faster with O₃ than isoprene. The rainforest boundary layer is a low NO_x environment, where photochemical O₃ production is limited by extremely low NO_x.”

I suggest the authors make numerical experiments changing the NO_x availability in the forested regions. The ozone variation here is moderated by the NO_x concentration not BVOCs.

Reply. We refer the reviewer to our reply above. We appreciate the reviewer’s comment highlighting that ozone production over the rainforest is limited by extremely low NO_x. However, we would like to clarify that in this section, we are not discussing ozone production. Instead, we are focusing on explaining the increase in surface-level O₃ concentrations when BVOCs are reduced. This increase can be attributed to the reduced loss of ozone due to a direct reaction with BVOC, namely ozonolysis. In the Amazon rainforest's boundary layer, where the primary sink for ozone is its reaction with BVOCs, a reduction in BVOCs results in less ozone being consumed, thereby leading to higher O₃ concentrations near the surface. Overall, this result indicates that in the lower troposphere, BVOCs act as a sink for ozone.

Besides the area immediately downwind of Manaus, the Amazon rainforest boundary layer contains very low NO_x. Indeed, the urban area cannot be adequately resolved by the resolution of this numerical simulation. Since soils in the rainforest region are not artificially fertilized (which would give rise to enhanced NO_x emissions), there is no rationale to examine the chemistry at high NO_x. Only in the case of widespread biomass burning would higher NO_x levels prevail over large areas. Since we saw no evidence of biomass burning (besides 4 short-duration spikes) during our 2-month campaign (as evidenced by acetonitrile), we consider a NO_x sensitivity study for the region to be beyond the scope of this paper.

We have added this point in the updated manuscript (Line no. 340-348)-

“The O₃ increase in the lower troposphere can be attributed to lower rates of direct BVOC-O₃ reactions, particularly for monoterpenes that react faster with O₃ than isoprene⁶². Therefore, we infer that BVOCs act mainly as the sink for O₃ (Supplementary Figure 14) because the rainforest boundary layer is a NO_x poor environment, where photochemical O₃ production is limited by extremely low NO_x^{50,51}. In contrast, the high-altitude regions are enriched in NO_x by

lightning, and the reduction in O_3 in the UT due to the reduction in BVOC levels reveals that O_3 production is limited by BVOCs, and that lower surface emissions mean less BVOC is available aloft to drive photochemical O_3 production.”

Figure R18. The flight tracks during the CAFE-Brazil campaign over the Amazon rainforest color-coded by Acetonitrile mixing ratios. Satellite picture data in this figure obtained from <https://wvs.earthdata.nasa.gov>, NASA Worldview Snapshots.

- Line 309-311 “We also estimate the radiative forcing changes over the Amazon region (calculated at the ATTO location) associated with the reduction of Amazon BVOC emissions by 50 and 75%, specifically due to the changes in O_3 and aerosol”

The authors estimated the radiative forcing based in only one data point (ATTO tower location) in the Amazon region, which does not represent the whole Amazon region. I recommend the authors to make this clearer.

Reply. We thank the reviewer for this suggestion. Indeed, we estimated the radiative forcing and also the changes in ozone and OH in Figure 5a, b at only one data point (ATTO). This was an unnecessary shortcoming of the submitted manuscript, and we have now changed this to cover the Amazon basin in the revised manuscript. We also made this clear in the text.

The revised sentence in the updated manuscript (Line no. 311-314)-

“The annual and seasonal changes in the Amazonian region over 0-20 km altitude, relative to the base case, for the oxidants OH radicals, O_3 , and the global radiative effect due to aerosol, are illustrated in Figure 5 and Supplementary Figure 13, respectively. Each panel in Figure 5 indicates the relative change across the atmospheric column.”

- Have the authors compared O_3 and aerosol observational data in the ATTO region with modeling result, before analyzing the changes due to the BVOCs emissions reductions (50% and 75%)? To claim that the radiative forcing changes over the Amazon region (because the reduction of Amazon BVOC emissions) are due changes in O_3 and aerosol its necessary to ensure the control model results agree with aircraft measurements. Shrivastava (2019) and Nascimento (2022) compare model and aircraft measurements by sampling the model data at the aircraft’s time and position to make a direct comparison. See the reference below:

Shrivastava, M.; Andreae, M. O.; Artaxo, P.; Barbosa, H. M. J.; Et al Urban pollution greatly enhances formation of natural aerosols over the Amazon rainforest. *Nat. Commun.* 2019, 10, 1046.

Reply. Unfortunately, observed ozone, and aerosol concentrations at the ATTO region are not available for this study. However, we refer the reviewer to our reply for the initial comment, where we evaluate ozone, OH and organic aerosol with observations from the ATom campaign (Figure R2-R4) and point to additional references for the evaluation of the EMAC model. Specifically for ATTO, we have however compared observations and EMAC numerical results for Black Carbon concentrations for the period 2019-2021 (Holanda et al. 2022)¹⁹.

- I recommend the authors to consider the difference in VOC concentrations in the Amazon region depending on the season. Concentrations of isoprene (m/z 69) and its oxidation products (m/z 71), biogenic precursors of OA, are a factor of 2–3 higher in the dry season than the wet season. See the following two references:

Gu, D., Guenther, A. B., Shilling, J. E., Yu, H., Huang, M., Zhao, C., Yang, Q., Martin, S. T., Artaxo, P., Kim, S., Seco, R., Stavrou, T., Longo, K. M., Tóta, J., de Souza, R. A. F., Vega, O., Liu, Y., Shrivastava, M., Alves, E. G., Santos, F. C., Leng, G., and Hu, Z.: Airborne observations reveal elevational gradient in tropical forest isoprene emissions, *Nat. Commun.*, 8, 1–7, <https://doi.org/10.1038/ncomms15541>, 2017.

Other references recommended to the authors:

Yáñez-Serrano, A. M.; Bourtsoukidis, E.; Alves, E. G.; Bauwens, M.; Stavrou, T.; Llusà, J.; Filella, I.; Guenther, A.; Williams, J.; Artaxo, P.; Sindelarova, K.; Doubalova, J.; Kesselmeier, J.; Peñuelas, J. Amazonian biogenic volatile organic compounds under global change. *Global Change Biol.* 2020, 26, 4722–4751.

Yáñez-Serrano, A. M. et al. Diel and seasonal changes of biogenic volatile organic compounds within and above an Amazonian rainforest. *Atmos. Chem. Phys.* 15, 3359–3378, doi:10.5194/acp-15-3359-2015 (2015).

Reply. We appreciate the reviewer's suggestion. However, our combined aircraft and ground-based dataset includes only the dry-to-wet transition period and wet-season, particularly for the upper troposphere. The GoAmazon 2014/15 campaign covered a much smaller area, lower down, primarily near Manaus, which is not representative of the pristine Amazon rainforest. Yáñez-Serrano et al. presented a ground-based study of BVOCs near the ATTO tower for both wet and dry seasons^{9,20}. However, comparing this to upper tropospheric data is challenging due to the limited convection activity during the dry season compared to the wet season. As Machado et al. (2004)¹⁰ reported, "The transition season from dry to wet and the beginning of the wet season is generally the period of strongest intensity of convection." That was the reason for planning this CAFE- Brazil campaign during December and January (transition season from dry to wet) to investigate the impact of convection on trace gases transport to the upper troposphere and their role in new particle formation.

Nevertheless, as per the reviewer's suggestion, we investigated the sensitivity of BVOCs emissions by applying a 50% increase and 50% and 75% reductions. In this sensitivity study, we have also included the seasonal variation of O₃, OH and the global radiative effect due to aerosol over the Amazon rainforest (see Figure R1 and R6).

Response to Reviewer #2:

The authors report surface and aircraft measurements of biogenic volatile organic compounds and their oxidation products, collected over the Amazon forest. They also use a global 3-D atmospheric chemistry model to interpret these data and to explore the sensitivity of their results to future deforestation and the corresponding changes in radiative forcing. This reviewer enjoyed the paper and found it scientifically interesting, but found some of the later modelling experiments to be an unnecessary extension of the observational results. I have mostly described very minor comments.

The manuscript is well written but the authors should take pity on this reviewer's eyes and make the figures bigger. Figures 2-4 are too small. The inset figure in Figure 2 is virtually unreadable – given the space in the main figure the author could afford to make the inset figure bigger.

We wish to thank the reviewer for thorough evaluation of our manuscript and valuable comments. The manuscript was revised by incorporating the comments and a point-by-point response is given below. The modelling experiments have been refined to better align with the main observational results. We have also modified figures 2-4 as per reviewer's suggestion.

Line 131: How did the authors estimate 20% of isoprene escapes the BL?

Reply. It is calculated by the average value measured in the upper troposphere to average values measured in the boundary layer during the whole campaign. This provides an overview of how much these BVOCs are transported from the boundary layer to the upper troposphere due to convection.

Line 139: Fussy, I know, but I do wonder whether “innate” is needed.

Reply. We have removed this word “innate” in the revised manuscript (**Line no. 164-166**).

“These processes also contribute to the large variability of BVOC mixing ratios observed in the UT, and the stochastic nature of the convective events.”

Line 171: The NASA Atmospheric Tomography (ATom) Mission should be introduced. Not ATOM.

Reply. We have corrected it in the revised manuscript (**Line no. 196-198**).

“Curtius et al. (2024)¹⁶ have reported extremely high particle production rates for a convective system, and the NASA Atmospheric Tomography (ATom) Mission have reported unexplained high particle numbers emerging from the Amazon region^{35,43,44}.”

Line 192: If you're using the 24-hour clock, please use 06:00 not 6:00. Here and elsewhere in the manuscript.

Reply. We have modified it everywhere in the revised manuscript.

Line 220: Awkward sentence using parentheses.

Reply. We have removed it in the revised manuscript. The revised sentence is

(Line no. 246-249) -

“The mean mixing ratio of monoterpenes in the UT at this time is closely comparable to those measured in the boundary layer at 320 m between 04:00 and 06:00 local time, suggesting that very efficient nighttime vertical transport has occurred.”

Line 230: Is it that remarkable? If you built a Hovmoller plot, would you see this delay is associated with the vertical transport of the material? Does ECHAM/MESSy reproduce this result?

Reply. Thank you for your suggestion. It would be really interesting to see the delay in the vertical transport of BVOCs from the boundary layer, but unfortunately, we did not fly low in the boundary layer during nighttime (not allowed by VFR flight rules in the dark), so we don't have enough aircraft data to see this delay.

However, we investigated this delayed transport using the EMAC model. Although the comparison of observed and simulated profiles of monoterpenes and isoprene indicates that the model underestimates UT BVOC levels in direct comparisons, the maximum simulated isoprene mixing ratios over the Amazon basin align well with the maximum observed values. This can be attributed to the spatially averaging of the EMAC model over each grid box and the specific targeting of outflow events during the CAFE-Brazil campaign.

For that reason, we show a Hovmoller plot of the diel cycle of maximum simulated isoprene mixing ratios over the Amazon rainforest, averaged over December 2022 and January 2023 (Figure R19). Indeed, the observed diel cycle and delayed transport can be reproduced. The highest surface mixing ratios are simulated in the afternoon, with the residual layer forming due to shallow convection in the evening and night. In the early morning hours, we simulate the highest isoprene mixing ratios in the UT. In addition, we provide the changes to this scenario in the sensitivity studies. The interesting result is that isoprene mixing ratios react non-linearly to changes in BVOC emissions. Reducing BVOC emissions by 75% leads to a reduction of isoprene of at least 85% everywhere, and to an almost complete removal of isoprene in the UT at daytime. This can be attributed to the increased availability of OH due to the decreased reaction with BVOCs, which additionally depletes isoprene. Conversely, when increasing BVOC emissions by 50%, isoprene mixing ratios increase by 100-250% due to the depletion of OH, thereby reducing oxidation, and extending isoprene lifetime.

Figure R19. a) Simulated average diel cycle of maximum isoprene mixing ratios between December 2022 and January 2023 over the Amazon basin from the surface to the upper troposphere (UT). The diurnal profiles of relative changes in isoprene under simulations with b) 50%, c) 75% reduction and d) a 50 % increase in the biogenic volatile organic compounds

(BVOCs) emission. Highest boundary layer mixing ratios occur in the afternoon, and the highest UT isoprene mixing ratios in the early morning. The sensitivity simulations show a non-linear decrease (increase) in the UT, especially in the daytime, in response to decreased (increased) BVOC emissions.

Line 246: Can the author back this statement up with a model calculation?

Reply. Unfortunately, we cannot back this statement up with model calculations. We did not perform a deforestation study (only a BVOC sensitivity study), and we only have grid boxes of ~200×200 km available.

Line 277: This reviewer would argue that this model error is associated more with sub-grid scale vertical motion rather than emissions averaged over large areas. Otherwise, you wouldn't have such good agreement in the BL and LT. Worth some additional discussion.

Reply. We agree with the reviewer and regret the incorrect formulation. The discrepancy (not necessarily model error) in the UT between the model and observations is because of the actively sought convective outflow during the CAFE-Brazil flights, which is indeed represented by sub-grid vertical motion. In the model, this vertical motion is parameterized in ~200×200 km grid boxes, and we only account for average upward mass flux in these grid boxes. Additionally, the comparison has been performed by sampling the model along the flight tracks, and convective events in the model do not necessarily coincide with the observed convective events. We reformulated this in the manuscript. We show in the Hovmoller plot that, in general, we reach the same maximum UT isoprene mixing ratios as observed when sampling over the complete Amazon basin.

The revised sentences are (**Line no. 299-302**) -

“The measurements show higher BVOC levels in the UT than the model, which can be expected since the aircraft actively sought convective outflow. These sub-grid scale processes are parametrized in the model and averaged over the horizontal extent of the grid box.”

The authors then launch into a series of sensitivity numerical experiments, all of which are interesting although this reviewer suggests they could be more aligned with the main points of the paper associated with the data. At the very least, the authors would need to describe the elevated levels of NO_x associated with biomass burning associated with deforestation, etc.

Reply. Thank you for your suggestion. We understand the impression arises that the model studies are not aligned with the main points of the paper. We improved this alignment by the following adjustments:

- Now we made it clear in the updated manuscript that the model studies are not a deforestation study but a sensitivity study of varying BVOC emissions, considering a deforestation and a CO₂-doubling scenario. We included an additional sensitivity simulation, increasing the BVOC emissions by 50%, and removed the global reduction simulation.
- We used the model to more thoroughly investigate the vertical transport reported in the observational section, and studied its sensitivity to BVOC emissions. We find that the diel cycle in the different tropospheric layers can be well reproduced by the model, justifying these studies (see Figure R19).

- We explained more clearly, why and how we perform the sensitivity studies in the manuscript to avoid the impression of two unaligned studies.

Finally, also discussed above, our model study was not meant as a deforestation study; rather it is a BVOC sensitivity study. For that reason, we believe that the inclusion of elevated NO_x levels from biomass burning is not in the scope of the presented manuscript, in particular because biomass burning was a near-negligible influence during the field campaign.

Line 332: The authors felt obliged to provide a caveat about any inadvertent recommendation to the reader about cutting down the Amazon rainforest, and provided some of the competing impacts associated with such an idea. This reviewer suggests that if they keep this paragraph they should describe all the consequences not just carbon, but also water, albedo, etc. These are covered in more detail in Weber et al: <https://www.science.org/doi/10.1126/science.adg6196>.

Reply. We wanted to make the point that in addition to the BVOC-induced radiative changes revealed in our sensitivity study, any real-world forest reduction will occur through burning, which puts additional CO₂ into the system. We now make this point and refer the reader to more detailed deforestation studies included in the reference provided by the reviewer.

The revised paragraph is (Line no. 376-392)

“This study highlights the role of convection in shaping the atmospheric distribution of BVOCs and their contribution to upper tropospheric chemistry, including O₃ production, and OH loss. Given that reducing BVOC emissions leads to a slight reduction in radiative forcing due to O₃, and a concomitant shorter methane lifetime. It may be tempting to advocate future measures that reduce BVOC emissions to ameliorate global warming. However, the radiative warming forcing that occurs when monoterpene-generated aerosol is reduced overwhelms the small (5%) changes from the methane lifetime change. It should also be noted that the Amazon rainforest stores approximately 108 Pg C in living aboveground biomass^{25,64}. According to this study, if deforestation through burning accounts for the 50% reduction in Amazonian biomass, it would additionally release approximately 54 Pg C (equivalent to a global increase of 25 ppm CO₂) into the atmosphere, leading to a strong positive radiative forcing and warming, as well as a long-term perturbation of the global hydrological and carbon cycles. The potential effects of deforestation and afforestation on the Earth’s atmosphere are further discussed elsewhere, e.g., in Weber et al. (2024)⁶⁵. Our BVOC sensitivity study highlights the role these species play in the atmosphere. In short, by storing carbon and releasing BVOCs, the Amazon rainforest modulates atmospheric CO₂ concentrations and regulates oxidation processes in the upper and lower troposphere.”

Response to Reviewer #3:

Summary/ General comments:

The paper starts by discussing CAFE-Brazil measurement flights conducted between December 2022-January 2023 over the Amazonian rain forest. They display PTR-TOF measurements of isoprene, monoterpenes and some isoprene oxidation products over a large scale of different altitudes. The vertical profiles presented in this paper shows an enrichment of all three groups in the upper troposphere and that this enrichment happens specifically during the night due to reduced photochemical sink. Hereafter, the focus shifts to two flights performed over regions, where deforestation has taken place previously. The authors discuss two individual flights performed within the boundary layer and describe that monoterpene, isoprene and isoprene oxidation products have lower mixing ratios there as compared to the forested regions. The authors hereafter say that this will also reduce the availability of BVOCs to be transported to the upper troposphere. This leads to the final part of the paper, where O₃, OH and aerosol radiative forcing sensitivities to decreasing BVOC emissions is studied using the EMAC global atmospheric chemistry model, where the BVOC emissions have been either decreased by 50% or 75% over the Amazon or 50% globally. The results show that reducing BVOC emissions influences the oxidant budget in different ways depending on the altitude and the oxidant. Isoprene emission reductions play a key role in influencing the oxidant budget, further decreasing methane lifetime. Decreased emissions of monoterpenes reduce the aerosol radiative effect causing a positive radiative forcing.

Overall, this study comes across to me as a combination of two studies that do not link together perfectly. The deforestation part of the work is interesting, but it would have been better if the authors had addressed also the likelihood of deep convection even taking place over the deforested areas (see e.g., Wang et al., 2009), or shown data on the upper troposphere in the vicinity of these regions. Further on in the modeling part of the work, it would have been interesting if the convective transport of isoprene, isoprene OP and monoterpenes was accounted for in the model in a more sophisticated way, or the effect of deforestation on the likelihood of convection and vertical transport were examined. This would have brought more new insights into the work, as now the rather simple sensitivity study does not necessarily add much to Scott et al. (2018), who did a more comprehensive deforestation study (my opinion).

The paper is interesting and timely due to the high interest of the scientific community on UT NPF and convective transport of isoprene. While not discussed in the paper, the observed elevated mixing ratios of isoprene and some oxidation products are in line with previous research in areas with convective activity (e.g., previous flight campaigns in Amazon and South-Eastern United States) as airborne observations using the PTR-TOF are not unique to this study. Nonetheless, these findings have not been discussed much in the past. What is really remarkable in this work, is that sufficient data exists to derive diel cycles for the three groups that highlight the importance of nocturnal transport of the species to the upper troposphere due to less efficient photochemical sink.

Reply. Thank you for your comments and valuable suggestion. We have revised the manuscript considering all the comments. We have also modified the modeling part and included the comparison of isoprene and monoterpenes model data with observation at the ATTO tower, better aligning the different parts of the study.

In addition, we use the model to investigate the convective transport more thoroughly, reproducing the observed diel cycle, and studying the sensitivity of this transport to changes in BVOC emissions (see Figure R19). Unfortunately, the EMAC model is not suitable for studies on the impact of deforestation on convective events, and a regional model with higher resolution and land cover feedback on the dynamics would be needed. However, we believe that this is not in the scope of the manuscript, as we do not perform deforestation but a sensitivity study on BVOC emissions, which is expressed more adequately in the revised manuscript.

The point-point reply of each comment is given below, which also has been incorporated in the revised manuscript.

Comments:

1. The Supplementary Fig. 3 shows the observed and simulated isoprene and monoterpene concentrations. You state on L273–274 that they generally agree. Does this mean that this model would not need improvement in terms of the convective transport of isoprene, isoprene oxidation products or monoterpenes? I would personally not say that these agree well (if you plot with x-axis in linear scale). Would be interesting to see how much this difference influences your OH, O₃ budgets or aerosol radiative effect.

Reply. We agree with the reviewer that especially monoterpenes do not agree well with observational data in the lower troposphere. This is partly due to an overestimation of the removal of monoterpenes. Another reason for this discrepancy could be the simulated shallow convection in the model, which is rarely targeted in the observations, as no boundary layer flights have been performed during the night.

Both isoprene and monoterpenes are showing disagreement in the upper troposphere. This discrepancy can be attributed to the actively sought convective outflow during the CAFE-Brazil flights, which is represented by sub-grid vertical motion. In the model, this vertical motion is parameterized in ~200×200 km grid boxes, and we only account for average vertical mass fluxes in these grid boxes. Additionally, the comparison has been performed by sampling the model along the flight tracks, and convective events in the model do not necessarily coincide exactly with the observed convective events (see also the reply to reviewer 2 above). We present the comparison on a logarithmic scale, as otherwise, barely anything can be seen in the UT. We also discuss the mentioned discrepancies in more detail in the revised manuscript.

In addition, we provide a Hovmoller plot of maximum simulated values over the Amazon rainforest, showing that we can reproduce the highest observed isoprene mixing ratios in the UT (Figure R19).

The revised sentences in the updated manuscript are given below (Line no. 294-306)

“We performed a base-case scenario and three sensitivity runs with perturbed BVOC emissions, as stated above. The base-case EMAC run generally agreed with the isoprene

measured during the CAFE-Brazil campaign, especially in the LT (see Supplementary Figure 8). While monoterpenes are somewhat underestimated in the boundary layer, most probably due to their reactions with O₃, which is known to be overestimated in EMAC, especially in the tropics⁵⁵, as well as in other global chemistry-climate models⁵⁶. The measurements show higher BVOC levels in the UT than the model, which can be expected since the aircraft actively sought convective outflow. These sub-grid scale processes are parametrized in the model and averaged over the horizontal extent of the grid box. However, surface monoterpene and isoprene mixing ratios were evaluated with measurements from the ATTO tower, and simulated ozone, OH and organic aerosol were additionally evaluated with background aircraft observations, showing good agreement (supplementary Figures 9-12). Moreover, previous studies have evaluated the EMAC model with similar setups for aerosols⁵⁷⁻⁶⁰, ozone and OH⁵⁵.”

2. How did you identify deep convection? Could you show some evidence that the observed elevated UT mixing ratios are associated with it? Could you also specify which observations you refer to on L129-131? Is it isoprene?

Reply. Convection is the vertical movement of air, driven by temperature differences, a key process for transferring heat, moisture and trace compounds throughout the depth of the troposphere. In this study, the deep convection events were identified based on the observed elevated levels of short-lived primary BVOCs, such as isoprene and monoterpenes, in the upper troposphere (see Figure R7). These compounds, emitted from terrestrial sources, typically have atmospheric lifetimes of 1-2 hours, making their transport to such high altitudes unlikely without deep convective activity. Meteorological data shows clear evidence of large-scale convection over the entire Amazon basin, for example, the cloud top brightness temperature variation (see Figure R8). A very low cloud-top brightness temperature (often below -50°C to -70°C) indicates deep convection.

To address this, we have revised the manuscript to include a figure illustrating a specific flight where high isoprene mixing ratios were measured in the upper troposphere, providing direct evidence of convective transport events (See Figure R7). We have also included the cloud top brightness temperature to identify the convective system and HALO aircraft location with high isoprene values (Figure R8).

Yes, it is isoprene and we added the new figure and the sentences in the updated manuscript as given below

(Line no. 144-147)

“The elevated mixing ratios of isoprene, isoprene-OP and monoterpenes in the UT provide clear evidence of extensive rapid transport to this altitude associated with deep convection. Supplementary Figure 3 depicts an example of the deep convection of isoprene in the UT for Flight 10.”

(Line no. 152-155)

“Gettelman and Forster³⁶ reported a lapse rate minimum height of 10-12 km for active convective (deep convection) regions in the tropics, in good agreement with the observed higher BVOCs at this altitude range presented here.”

3. Many previously published studies could be highlighted:

a. The results from Fig. 2 for isoprene are in line with previously published data from GoAmazon (Shilling et al., 2018; Jo et al., 2021) and SEARC4s from south-eastern US (Jo et al., 2021), which also experience convective activity. There are surely many more.

Reply. Thank you for your suggestions. We have included the reference Jo et al. (2021)²¹ in the revised manuscript for the SEARC4s campaign over the southeastern US. While Shilling et al. (2018)¹³ and Jo et al. (2021)²¹ do not specifically discuss elevated isoprene levels in the upper troposphere for the GoAmazon campaign, we have cited these references in the introduction section to provide additional context. Due to the journal's limitation on the number of references, we are unable to include a larger number of citations. We hope the reviewer appreciates this constraint and our effort to prioritize the most relevant studies for this work.

(Line no. 158-160)

“Jo et al. (2021)³⁷ measured on average ~25% of isoprene from the boundary layer was transported to the UT during the SEARC4RS campaign over the southeastern US.”

b. L170–L172: The high particle number concentrations in amazon outflow in the UT, associated with a convective system and isoprene oxidation products, were shown by Zha et al. (2024). The Curtius (2024) paper is not provided within this review file and it is not published, so I cannot evaluate this sentence.

Reply. We have added the reference “Zha et al. (2024)²²” as per the reviewers’ suggestion. Curtius et al. (2024)²³ is published now so we have included the proper reference for this paper in the reference section of the revised manuscript **(Line no. 196-198)**

“Curtius et al. (2024)¹⁶ have reported extremely high particle production rates for a convective system, and the NASA Atmospheric Tomography (ATom) Mission have reported unexplained high particle numbers emerging from the Amazon region^{35,43,44}.”

4. Fig. 2 shows the 1 min-averaged data and the 1km bin means. Wouldn't it make more sense to show a median and the interquartile range?

Reply. In Figure 2, we present 1-minute averaged data points (grey dots), as the collected data has a 1-second time resolution. The green line represents the mean values of these 1-minute data points for each 1-km bin. We agree with the reviewer to add a median and the interquartile range, but it would complicate the figure unduly, and the main objective of this figure is to show the vertical profile and the transport of BVOCs in the UT due to convection. We wanted to show the comparison in the level of BVOCs in the UT during the day and nighttime. However, as per the reviewer's suggestion, we included the box whisker plot with mean and median values for the whole data, nighttime and daytime data in the supplementary information (Figure R13 & Figure R20).

The green lines do not really seem representative of the 1 min data. How are the day and night plots made? Could you also show the 1 min data for both and make two panels?

Reply. In Figure 2, the green line is the mean values of biogenic volatile organic compounds (BVOCs) for 1 km bin and the mean value is calculated for the values above limit of detection (LOD). While the grey dots represent the all-data points, including below the detection limit, to show the background values as well. We have added the box-whisker plot for all data, as

well as the day and nighttime vertical profile of BVOCs in the supplementary information (Figure R13 & Figure R20).

Figure R20. Vertical profiles of isoprene, its oxidation products (OP), and monoterpenes during the CAFE-Brazil campaign over the Amazon rainforest (15 days of data= \sim 125hr). The profiles are binned to a 1 km vertical resolution grid, with the number of data points used for each box displayed above it. The boxes represent the 25th–75th percentiles, with black lines indicating the median. Whiskers extend to the lowest and highest data points within $1.5\times$ Interquartile Range (IQR), the data point beyond this range is considered outlier (black dots). Only data above the limit of detection (LOD) is used for this box-whisker plot. The number of data points for these biogenic volatile organic compounds (BVOCs) differs due to their respective LOD values.

5. It is unclear how the BVOC emission reductions are done for the model simulations. I'm confused by L280-L281– is this just unnecessary repetition? Could you provide a more comprehensive description of the model and the simulations? I think you should also be careful with direct comparisons to Scott et al (2018) or state that what you are studying is not the impact of deforestation.

Reply. The emission reductions are applied by scaling the emission fluxes by the relevant factors. We understand the confusion about L280-L281 and regret the unclear formulation. In summary, we wanted to show the reduction in isoprene mixing ratios in response to reduction in isoprene emissions, that not necessarily have to be linear. We have now removed this sentence and provide more information on the simulation and the sensitivity study in the Methods section.

For clarity, we now state that we are conducting a sensitivity study and not a deforestation study, and we now include both an increase and decrease in BVOC to emphasize this difference. The only connection to deforestation is that we lower the model emissions in the sensitivity study to the approximate levels measured over forest-cleared pasture.

The revised paragraph in the updated manuscript is – **(Line no. 263-289)**

“The EMAC model simulations were performed to investigate the sensitivity of atmospheric chemistry above the Amazonia basin and the radiative effects to changes in BVOC emissions. The motivation of the sensitivity study is the elevated levels of BVOCs measured in the UT

associated with convection and addressing how changes in the BVOC emission at the surface will impact the atmospheric chemistry from the boundary layer to the UT (9-14 km). Flight 06 provides notable motivation for this sensitivity study. In this flight, in fact, the measurements documented BVOCs over deforested areas converted to pasture, and contrasted to nearby pristine forests sampled at relatively low altitudes (~900 m) in the boundary layer over Rondônia, Brazil. Figures 4a-d depict the variations in the altitude and mixing ratios of isoprene, isoprene-OP and monoterpenes along flight 6 over forested and deforested regions, respectively.

The flight 06 data show a steep spatial gradient of isoprene mixing ratio over forested and pasture regions. The mean mixing ratios of isoprene, isoprene-OP, and monoterpenes over deforested regions were 0.75 ± 0.45 , 1.2 ± 0.36 and 0.08 ± 0.04 ppbv, respectively. In contrast, the mean mixing ratios of isoprene (2.96 ± 0.72 ppbv), and monoterpenes (0.31 ± 0.09 ppbv) over forested regions were approximately four times higher than the pasture regions. While mixing ratio of isoprene-OP (2.12 ± 0.41 ppbv) were only ~1.7 times higher than over the deforested regions. The lower spatial gradients of isoprene-OP are likely due to the transport of the longer-lived isoprene-OP from adjacent forested regions.

Deforestation has been reported as a primary driver of the reduction in BVOC emissions in tropical forests although other factors, such as biomass burning and climate change, also play significant roles. On the other hand, several studies suggested that global BVOC emissions will likely increase with climate change, potentially by up to 50% under scenarios with doubled CO₂ concentrations^{52,53}. As a result, the future Amazonian atmosphere could experience dramatic positive or negative variations in BVOC emissions. Therefore, we evaluate the atmospheric response in scenarios where Amazonian BVOC emissions are reduced by 50% and 75%, and increased by 50% using the EMAC model.”

The included sentence in the method section of the updated manuscript is – **(Line no. 468-470)**

“For the sensitivity studies, BVOC emissions were simply scaled with a factor of 0.5 (50% reduction), 0.25 (75% reduction), and 1.5 (50% increase) without further changes to the simulation.”

6. Isn't aerosol radiative effect more accurate term for this kind of study than aerosol radiative forcing?

Reply. We agree with the reviewer that the aerosol radiative effect is more accurate, and we changed it accordingly in the manuscript.

7. How is the global aerosol RF calculated? How can you have a vertically resolved W m⁻² when RF should be for the full column? Did you “slice” this somehow? It is also confusing in the text when you talk about “global radiative forcing” if you mean “global aerosol radiative forcing”.

Reply. We regret the confusion, especially the wrong reference to “global radiative forcing”. We corrected that in the revised manuscript. The radiative forcing or rather the radiative effect, is the imbalance of incoming and outgoing radiation. This is normally calculated at the top of the atmosphere or at the surface. However, this can also be calculated at every vertical level. We present the imbalance occurring due to the changes in BVOC emissions. We state this more clearly in the caption now.

Line no. 350-353

“The vertical profiles of relative changes in a) O₃ and b) OH and of absolute changes in c) the global radiative effect due to aerosol between the base simulation and the simulation with 50% and 75% reduction and a 50 % increase in the BVOCs emission. The radiative effect is the imbalance between incoming and outgoing radiation and is calculated at each vertical level”

Minor comments:

8. Some figures don't have labels, perhaps add the missing?

Reply. We have revised all figures with better resolution and rechecked the legends and axis labels as well.

9. Figure 3: You could also specify that 320m refers to ATTO in the legend for clarity. In the legend, it is hard to distinguish green from blue on my screen at least (plot lines are clear though). Consider increasing linewidth for the legend.

Reply. We have revised Figure 3 to make it clearer. On the x-axis, we added 'ATTO (at 320 m)' to clarify that the diurnal profile for ATTO corresponds to 320 m. Additionally, we updated the legend style and used colors in the legend to represent the altitudes included in the diurnal profile (Figure R14).

10. L105: chemical oxidation sounds a bit odd, maybe change to chemical loss (oxidation)?

Reply. We replaced the “chemical oxidation” with “chemical loss (oxidation)”. The revised sentence in the updated manuscript (**Line no. 125-127**)-

“Within the LT, mixing ratios decrease rapidly with height through both turbulent mixing dilution from above and chemical loss (oxidation).”

11. Citation 11 seems to be wrong. Perhaps you refer to Williams et al. (2000): An Atmospheric Chemistry Interpretation of Mass Scans Obtained from a Proton Transfer Mass Spectrometer Flown over the Tropical Rainforest of Surinam? The current cited article seems not to include any data on BVOCs.

Reply. We revised the citation, replaced by Williams et al. (2001)²⁴.

(Line no. 37-39)

“Indeed, a strong concentration gradient has been observed by day at the entrainment zone between the top of the planetary boundary layer (0-2.5 km) and the free tropical troposphere (2.5-18 km)¹¹.”

“Williams, J., Pöschl, U., Crutzen, P.J., Hansel, A., Holzinger, R., Warneke, C., Lindinger, W. and Lelieveld, J., 2001. An atmospheric chemistry interpretation of mass scans obtained from a proton transfer mass spectrometer flown over the tropical rainforest of Surinam. *Journal of Atmospheric Chemistry*, 38, pp.133-166.”

References

Jo, D. S., Hodzic, A., Emmons, L. K., Tilmes, S., Schwantes, R. H., Mills, M. J., Campuzano-Jost, P., Hu, W., Zaveri, R. A., Easter, R. C., Singh, B., Lu, Z., Schulz, C., Schneider, J., Shilling, J. E., Wisthaler, A., and Jimenez, J. L.: Future changes in isoprene-

epoxydiol-derived secondary organic aerosol (IEPOX SOA) under the Shared Socioeconomic Pathways: the importance of physicochemical dependency, *Atmos. Chem. Phys.*, 21, 3395–3425, <https://doi.org/10.5194/acp-21-3395-2021>, 2021.

Scott, C. E., Monks, S. A., Spracklen, D. V., Arnold, S. R., Forster, P. M., Rap, A., Äijälä, M., Artaxo, P., Carslaw, K. S., Chipperfield, M. P., Ehn, M., Gilardoni, S., Heikkinen, L., Kulmala, M., Petäjä, T., Reddington, C. L. S., Rizzo, L. V., Swietlicki, E., Vignati, E., and Wilson, C.: Impact on short-lived climate forcers increases projected warming due to deforestation, *Nat Commun*, 9, 157, <https://doi.org/10.1038/s41467-017-02412-4>, 2018.

Shilling, J. E., Pekour, M. S., Fortner, E. C., Artaxo, P., De Sá, S., Hubbe, J. M., Longo, K. M., Machado, L. A. T., Martin, S. T., Springston, S. R., Tomlinson, J., and Wang, J.: Aircraft observations of the chemical composition and aging of aerosol in the Manaus urban plume during GoAmazon 2014/5, *Atmos. Chem. Phys.*, 18, 10773–10797, <https://doi.org/10.5194/acp-18-10773-2018>, 2018.

Wang, J., Chagnon, F. J. F., Williams, E. R., Betts, A. K., Renno, N. O., Machado, L. A. T., Bisht, G., Knox, R., and Bras, R. L.: Impact of deforestation in the Amazon basin on cloud climatology, *Proceedings of the National Academy of Sciences*, 106, 3670–3674, <https://doi.org/10.1073/pnas.0810156106>, 2009.

Zha, Q., Aliaga, D., Krejci, R., Sinclair, V. A., Wu, C., Ciarelli, G., Scholz, W., Heikkinen, L., Partoll, E., Gramlich, Y., Huang, W., Leiminger, M., Enroth, J., Peräkylä, O., Cai, R., Chen, X., Koenig, A. M., Velarde, F., Moreno, I., Petäjä, T., Artaxo, P., Laj, P., Hansel, A., Carbone, S., Kulmala, M., Andrade, M., Worsnop, D., Mohr, C., and Bianchi, F.: Oxidized organic molecules in the tropical free troposphere over Amazonia, *National Science Review*, 11, nwad138, <https://doi.org/10.1093/nsr/nwad138>, 2024.

References-

1. Pozzer, A. *et al.* Distributions and regional budgets of aerosols and their precursors simulated with the EMAC chemistry-climate model. *Atmospheric Chemistry and Physics* **12**, 961–987 (2012).
2. Pozzer, A. *et al.* Simulation of organics in the atmosphere: evaluation of EMACv2. 54 with the Mainz Organic Mechanism (MOM) coupled to the ORACLE (v1. 0) submodel. *Geoscientific model development* **15**, 2673–2710 (2022).
3. Tsimpidi, A., Karydis, V., Pozzer, A., Pandis, S. & Lelieveld, J. ORACLE (v1. 0): module to simulate the organic aerosol composition and evolution in the atmosphere. *Geoscientific Model Development* **7**, 3153–3172 (2014).

4. Kohl, M. *et al.* Numerical simulation and evaluation of global ultrafine particle concentrations at the Earth's surface. *Atmospheric Chemistry and Physics* **23**, 13191–13215 (2023).
5. Jöckel, P. *et al.* Earth System Chemistry integrated Modelling (ESCiMo) with the Modular Earth Submodel System (MESSy) version 2.51. *Geoscientific Model Development* **9**, 1153–1200 (2016).
6. Elkins, J. W., Hints, E. J. & Moore, F. L. ATom: Measurements from the UAS Chromatograph for Atmospheric Trace Species (UCATS). (2019)
doi:10.3334/ORNLDAAAC/1750.
7. Brune, W. H. *et al.* Extreme oxidant amounts produced by lightning in storm clouds. *Science* **372**, 711–715 (2021).
8. Jimenez, J. *et al.* ATom: L2 Measurements from CU High-Resolution Aerosol Mass Spectrometer (HR-AMS), ORNL DAAC, Oak Ridge, Tennessee, USA. (2019).
9. Yáñez-Serrano, A. M. *et al.* Diel and seasonal changes of biogenic volatile organic compounds within and above an Amazonian rainforest. *Atmospheric Chemistry and Physics* **15**, 3359–3378 (2015).
10. Machado, L. A. T., Laurent, H., Dessay, N. & Miranda, I. Seasonal and diurnal variability of convection over the Amazonia: A comparison of different vegetation types and large scale forcing. *Theoretical and Applied Climatology* **78**, 61–77 (2004).
11. Andreae, M. O. *et al.* The Amazon Tall Tower Observatory (ATTO): overview of pilot measurements on ecosystem ecology, meteorology, trace gases, and aerosols. *Atmospheric Chemistry and Physics* **15**, 10723–10776 (2015).
12. Pöhlker, M. L. *et al.* Long-term observations of cloud condensation nuclei in the Amazon rain forest—Part 1: Aerosol size distribution, hygroscopicity, and new model

- parametrizations for CCN prediction. *Atmospheric Chemistry and Physics* **16**, 15709–15740 (2016).
13. Shilling, J. E. *et al.* Aircraft observations of the chemical composition and aging of aerosol in the Manaus urban plume during GoAmazon 2014/5. *Atmospheric Chemistry and Physics* **18**, 10773–10797 (2018).
 14. Ganzeveld, L. & Lelieveld, J. Impact of Amazonian deforestation on atmospheric chemistry. *Geophysical Research Letters* **31**, (2004).
 15. Weber, J. *et al.* Chemistry-driven changes strongly influence climate forcing from vegetation emissions. *Nature Communications* **13**, 7202 (2022).
 16. Atkinson, R. & Arey, J. Atmospheric degradation of volatile organic compounds. *Chemical reviews* **103**, 4605–4638 (2003).
 17. Hantschke, L. *et al.* Atmospheric photooxidation and ozonolysis of Δ^3 -carene and 3-caronaldehyde: rate constants and product yields. *Atmospheric Chemistry and Physics* **21**, 12665–12685 (2021).
 18. Oliveira, R. C. de M. & Bauerfeldt, G. F. Ozonolysis Reactions of Monoterpenes: A Variational Transition State Investigation. *J. Phys. Chem. A* **119**, 2802–2812 (2015).
 19. Holanda, B. A. *et al.* African biomass burning affects aerosol cycling over the Amazon. *Communications Earth & Environment* **4**, 154 (2023).
 20. Yáñez-Serrano, A. M. *et al.* Amazonian biogenic volatile organic compounds under global change. *Glob Chang Biol* **26**, 4722–4751 (2020).
 21. Jo, D. S. *et al.* Future changes in isoprene-epoxydiol-derived secondary organic aerosol (IEPOX SOA) under the Shared Socioeconomic Pathways: the importance of physicochemical dependency. *Atmospheric Chemistry and Physics* **21**, 3395–3425 (2021).

22. Zha, Q. *et al.* Oxidized organic molecules in the tropical free troposphere over Amazonia. *National Science Review* **11**, nwad138 (2023).
23. Curtius, J. *et al.* Isoprene nitrates drive new particle formation in Amazon's upper troposphere. *Nature* (2024).
24. Williams, J. *et al.* An Atmospheric Chemistry Interpretation of Mass Scans Obtained from a Proton Transfer Mass Spectrometer Flown over the Tropical Rainforest of Surinam. *Journal of Atmospheric Chemistry* **38**, 133–166 (2001).